# An Annotated Checklist of Monogeneans (Platyhelminthes, Monogenea) from Aquatic Vertebrates in Peru: A Review of Diversity, Hosts and Geographical Distribution

**DOI:** 10.3390/ani14111542

**Published:** 2024-05-23

**Authors:** Luis Angel Santillán, Celso Luis Cruces, Gloria M. Sáez, Rosa Martínez-Rojas, Aarón Mondragón-Martínez, Germán Augusto Murrieta Morey, Mauro Quiñones, José Luis Luque, Jhon Darly Chero

**Affiliations:** 1Laboratorio de Zoología de Invertebrados, Departamento Académico de Zoología, Facultad de Ciencias Biológicas, Universidad Nacional Mayor de San Marcos (UNMSM), Av. Universitaria Cruce con Av. Venezuela Cuadra 34, Lima 15081, Peru; luis.santillan5@unmsm.edu.pe; 2Facultad de Ciencias Biológicas, Universidad Ricardo Palma (URP), Av. Alfredo Benavides 5440 Santiago de Surco, Lima 15039, Peru; celso.cruces@urp.edu.pe (C.L.C.); mauro.quinones@urp.edu.pe (M.Q.); 3Laboratorio de Parasitología General y Especializada, Facultad de Ciencias Naturales y Matemática, Universidad Nacional Federico Villarreal (UNFV), Lima 15007, Peru; gsaez@unfv.edu.pe; 4Laboratorio de Parasitología de Fauna Silvestre y Zoonosis, Departamento Académico de Microbiología y Parasitología, Facultad de Ciencias Biológicas, Universidad Nacional Mayor de San Marcos (UNMSM), Av. Universitaria cruce con Av. Venezuela Cuadra 34, Lima 15081, Peru; rmartinezr@unmsm.edu.pe (R.M.-R.); aaron72.mondragon@gmail.com (A.M.-M.); 5Laboratorio de Parasitología y Sanidad Acuícola, Instituto de Investigaciones de la Amazonía Peruana (IIAP), Iquitos 16001, Peru; 6Programa de Pós-Graduação em Ciência Animal (PPGCA), Universidade Estadual do Maranhão (UEMA), São Luis 65055-970, Brazil; 7Departamento de Parasitologia Animal, Universidade Federal Rural do Rio de Janeiro, Seropédica 23890-000, Brazil; luqueufrrj@gmail.com

**Keywords:** species diversity, fish ectoparasites, amphibians, Dactylogyridae, Diclidophoridae, Diplectanidae, neotropical region

## Abstract

**Simple Summary:**

**Simple Summary:** Monogeneans are flatworm parasites that live principally in the gills of fish and sometimes infect other animals like amphibians, reptiles, and even mammals such as hippos. However, our understanding of these parasites in Peru is limited. To fill this gap, we compiled a detailed list of monogeneans found in Peru by studying the existing literature and examining specimens in collections. This list includes information about diversity, hosts, and geographical distribution. We reported 358 species of monogeneans, mostly infecting fish, with a few infecting amphibians. Most of these parasites live in fresh water, but some are found in marine environments. The most common families of monogeneans are Dactylogyridae and Diplectanidae. Some fish species harbor more parasites than others. However, many species have not been properly studied or collected, highlighting the need for further research to fully understand the diversity of monogeneans in Peruvian aquatic ecosystems.

**Abstract:**

Monogeneans are flatworm parasites that infest fish gills primarily but can also infect various other vertebrates, including amphibians, aquatic reptiles, mammals like hippos, and occasionally invertebrates like copepods, isopods, and cephalopods. Despite their remarkable diversity, our knowledge of monogenean parasites in Peru remains significantly limited, resulting in substantial gaps in our comprehension of their taxonomic identities, host associations, and geographic distribution. To address these knowledge deficits, we present an extensively curated checklist of monogeneans associated with aquatic vertebrates in Peru. This comprehensive compilation is derived from meticulous literature surveys, the examination of specimens deposited in both international and national collections, and the inclusion of additional freshly collected specimens. The checklist offers a thorough repository of data encompassing the diversity, host associations, and geographical distribution of these parasites. Taxonomic discrepancies are addressed through a critical review of the existing literature, supplemented by the direct examination of specimens, including type or voucher specimens, deposited within scientific collections. Additionally, we provide data on the DNA sequences of individual taxa. The compiled list comprises records of 358 monogenean species, including 270 valid species and 88 taxa identified at the family or generic level, all reported across 145 host species in Peru. Predominantly, these parasitic species exhibit associations within fish, with 335 infecting teleosts and 20 affecting chondrichthyans. Three monogenean species have been documented as infecting amphibians, namely *Mesopolystoma samiriensis*, *Polistoma* sp. and *Wetapolystoma almae*. Among the monogeneans reported, 141 were found in marine environments and 214 in freshwater environments. The most diverse families were Dactylogyridae and Diplectanidae, comprising 217 and 24 species, respectively. The hosts that harbored the highest number of monogeneans were *Pygocentrus nattereri* (with 23 species), followed by *Stellifer minor* (13 spp.) and *Triportheus angulatus* (11 spp.). We detected many species that do not have any material deposited in a scientific collection due to the loss or deactivation of the collection. These findings represent only a fraction of the potential diversity, considering the wide variety of aquatic vertebrate hosts inhabiting the tropical and subtropical regions of Peru.

## 1. Introduction

Monogeneans (Monogenea) constitute a group of flatworms characterized by the presence of a specialized attachment apparatus, known as a haptor, located at the posterior end of the body [1,2]. This taxonomic group encompasses a highly diverse array, primarily consisting of ectoparasites that infest the external surface, including the skin, fins, and gills, of both marine and freshwater fish [1]. Nonetheless, species belonging to the family Polystomatidae Gamble, 1896, infect internal organs like the cloaca or urinary bladder of vertebrate hosts, such as amphibians and aquatic reptiles [3]. Additionally, a subset of these organisms utilizes invertebrates, such as copepods, isopods, and cephalopods, as their direct hosts [4,5,6]. An exceptional case is the species *Oculotrema hippopotami* Stunkard, 1924, described as a parasite in the ocular cavity of hippopotamuses [7]. Given that monogeneans typically exhibit strict host specificity, they are regarded as suitable models for conducting studies on host–parasite coevolution [8].

According to Luque et al. [9], monogeneans represent the most species-rich group of platyhelminthes in Peru. Research on monogeneans from Peru dates back to the early 1970s [10]. The first monogenean species formally described in Peru was *Pseudoeurysorchis sarmientoi* Tantaleán, 1974. This diclidophorid species infects the gill filaments of *Seriolella violacea* [10,11]. Following that, Tantaleán [12,13] described *Cynoscionicola sciaenae* Tantaleán, 1974, *Choricotyle chimbotensis* Tantaleán, 1974, and *C*. *peruensis* Tantaleán, 1974. Numerous efforts have been undertaken in the past to assess the diversity of these flatworms as parasites in aquatic vertebrates from Peru [14,15,16]. While all of these checklists incorporate findings presented at scientific meetings, it is essential to note that any new records must fulfill specific criteria: they must be supported by deposited specimens in a scientific collection and undergo publication in a scientific journal. In a study conducted less than a decade ago, Luque et al. [9] compiled a comprehensive list of fish monogeneans, identifying a total of 175 species from Peru. Nonetheless, a detailed review of the Luque et al. [9] checklist revealed some discrepancies (i.e., mistaken records, lack of nomenclature updating, etc.). 

The main objective of this paper is to present an annotated checklist that succinctly compiles records of monogeneans in aquatic vertebrates from Peru while incorporating new data. This checklist provides comprehensive insights into their hosts, site of infection, geographical distribution, and genetic data. Additionally, it seeks to critically assess reports that introduce doubts or uncertainties. Ultimately, the objective of this study is to shed light on the existing obstacles that impede a thorough understanding of the diversity and host relationships of monogeneans in Peru.

## 2. Materials and Methods

The database of monogeneans associated with aquatic vertebrates in Peru was compiled using information from three primary sources: (1) an extensive analysis of literature published until November 2023, utilizing various databases (Helminthological Abstracts, Google Scholar, Science Direct, Web of Knowledge, Springer); (2) searches conducted within the database of the following international and national collections (MUSM-HEL, UNMSM-LPFSZ, USNM); and (3) bibliographical searches of articles published in Spanish that have not been indexed in electronic databases. Undergraduate theses and findings presented at scientific meetings were not included due to their lack of formal publication. The compiled monogenean-host list is presented in alphabetical order. Monogenea classification is based on WoRMS [17], and any nomenclatural changes are sourced from specific references mentioned in the notes section. The names of host species follow Froese and Pauly [18] for fish, and AmphibiaWEb [19] for amphibians. For each monogenean species, the database includes the scientific name, authority and year, host species, site of infection, locality, and corresponding reference. In the case of new species, the type locality, type host, and original reference are reported in brackets. In instances where a record is derived from a collection database but has not been published, the collection’s acronym is provided along with the record. Additionally, each record reports the acronym and accession number of the respective collections where the specimens were deposited. The acronyms used in the checklist are shown in Table 1.

## 3. Results

The database reported from the available literature on monogeneans in Peru comprises records of 270 species recognized as valid, as well as unidentified ones included in 74 genera and 14 families. These monogeneans are associated with 144 vertebrate host taxa, corresponding to 9 taxa of cartilaginous fish, 132 of bony fish, and 3 of amphibians. The records presented in this checklist account for 12 of the 24 Peruvian regions. The most sampled regions are Loreto and Lima, with 181 and 88 registered, respectively, whereas Arequipa and Amazonas have 2 and 3 records of monogeneans. In terms of diversity, the marine fish *Stellifer minor* Tschudi, 1846 and the freshwater fish *Pygocentrus nattereri* Kner, 1858, are the host species with the highest species richness across their distributional ranges; the former is distributed along the Peruvian coast and the latter in the Amazon River basin, with 13 and 23 monogenean taxa, respectively. *Anacanthorus* Mizelle & Price, 1965 and *Gussevia* Kohn & Paperna, 1964, with 24 and 21, respectively, are the genera with the largest number of species infesting freshwater fish in Peru. In the marine environment, *Rhamnocercus* Monaco, Wood & Mizelle, 1954 and *Hargicotyle* Mamaev, 1972, with 9 and 7, respectively, are the most diverse genera. 

### 3.1. Parasite–Host List 


**Phylum Platyhelminthes Gegenbaur, 1859**



**Class Monogenea Van Beneden, 1858**



**Subclass Monopisthocotylea Odhner, 1912**



**Order Capsalidea Lebedev, 1988**



**Family Capsalidae Baird, 1853**



***Benedenia* sp.**


*Isacia conceptionis*; gills; marine; La Libertad [20].

*Hyporthodus niphobles*; gills; marine; Lima (12°28′ S, 76°47′ W) (HPIA 144) [21].


***Benedeniella* sp.**


*Myliobatis peruvianus*; stomach; marine; Ica [16].

Remarks

No specimens in collections.


***Capsala biparasitica* (Goto, 1894) Price, 1938**


*Thunnus albacares*; nasal cavity; marine; Tumbes (03°29′ S, 80°24′ W) (MUSM-HEL 4720a-b) [22].

Remarks

Status: valid species.


***Capsala gregalis* (Wagner & Carter, 1967) Chisholm & Whittington, 2007**


*Sarda chiliensis*; gills, inner surface of the operculum; marine; Lima (12°30′ S, 76°50′ W) (CHURP 531) [23].

*Sarda chiliensis*; gills; marine; Lima (MUSM-HEL 1101) (unpublished data, MUSM-HEL). 

*Sarda chiliensis*; gills; marine; Tumbes (03°29′ S, 80°24′ W) (MUSM-HEL 4721a-e) [22].

Remarks

This species was recorded in Peru by Luque and Iannacone [23] as *Caballerocotyla australis* Oliva, 1986. However, the genus *Caballerocotyla* Price, 1960 was considered a synonym of *Capsala* Bosc, 1811 by Chisholm and Whittington [24]. Furthermore, the species is a synonym of *Capsala gregalis* (Wagner & Carter, 1967) Chisholm & Whittington, 2007. The voucher that was deposited in the CHURP is no longer available, as this collection has been deactivated. Status: valid species.


***Capsala paucispinosa* (Mamaev, 1968) Chisholm & Whittington, 2007**


*Thunnus obesus*; gills; marine; Piura (MUSM-HEL 1520) (new geographical record) (unpublished data, MUSM-HEL).

Remarks

Status: valid species.


**Capsalidae gen. sp.**


*Cheilodactylus variegatus*; gills; marine; Lima (UNMSM-LPFSZ 420) [25].

Remarks

This species was recorded as *Macrophyllida antarctica* (Hughes, 1928) Johnston, 1929 by Martínez et al. [25]. However, a taxonomic study of the voucher specimens deposited in the UNMSM-LPFSZ revealed that this record is not conspecific with *M*. *antarctica*, and actually represents a new genus.


***Encotyllabe antofagastensis* Sepúlveda, Gonzáles & Oliva, 2014**


*Anisotremus scapularis*; gills; marine; Lima (CPMP 071) [26].

Remarks

Status: valid species.


***Encotyllabe cheilodactyli* Sepúlveda, Gonzáles & Oliva, 2014**


*Cheilodactylus variegatus*; pharyngeal plates; marine; Lima (CPMP 500) (new geographical record) (present study).

*Cheilodactylus variegatus*; pharyngeal plates; marine; Piura (CPMP 501) (new geographical record) (present study).

Remarks

Status: valid species.


***Encotyllabe pagrosomi* MacCallum, 1917**


*Caulolatilus affinis*; gills; marine; Piura (04°13′ S, 81°13′ W) (UNMSM-LPFSZ 436) [27].

Remarks

Status: valid species.


***Encotyllabe* sp1.**


*Sciaena deliciosa*; gills, mouth; marine; Lima (03°29′ S, 80°24′ W) (MUSM-HEL 277) [28].

*Sciaena deliciosa*; gills, mouth; marine; Lima [16].

*Sciaena deliciosa*; gills, opercule; marine; Lima (CPMP 002) [29].

Remarks

These records were previously referred to as *Encotyllabe callaoensis* Tantaleán, 1973 by both Tantaleán [28] and Chero et al. [29]. However, *E*. *callaoensis* is considered a nomen nudum because it was described in an unpublished thesis and was never formally described in accordance with the rules of the International Code for Zoological Nomenclature (ICZN) [30]. Consequently, this monogenean needs to be formally described. 


**Encotyllabe sp2.**


*Stellifer minor*; gills; marine; La Libertad [31].

*Stellifer minor*; gills; marine; Lambayeque [16].

Remarks

No specimens in collections.


***Encotyllabe* sp3.**


*Aplodactylus punctatus*; gills; marine; Ica [16].

Remarks

No specimens in collections. 


***Listrocephalos kearni* Bullard, Payne & Braswell, 2004**


*Hypanus dipterurus*; skin; marine; Lima (12°09′ S, 77°01′ W) (MUSM-HEL 3243) [22].

Remarks

Status: valid species.

*Nasicola klawei* (Stunkard, 1962) Yamaguti, 1968

*Thunnus obesus*; nasal cavity; marine; Piura (05°04′ S, 81°06′ W) (UNMSM-LPFSZ 448) [32].

*Thunnus albacares*; nasal cavity; marine; Tumbes (03°29′ S, 80°24′ W) (MUSM-HEL 4722a-m) [22].

Remarks

Status: valid species.


***Neobenedenia pacifica* Bravo-Hollis, 1971**


*Mugil cephalus*; gills; marine; Lima (12°4′ S, 77°10′ W) (CPMP 502) (new geographical record) (present study).

Remarks

Status: valid species.


***Neobenedenia* sp.**


Opidiidae not identified; gills; marine; Lima [16].

Remarks

No specimens in collections.


***Sprostoniella lamothei* Pérez-Ponce de Léon & Mendoza-Garfias, 2000**


*Parapsettus panamensis*; gills; marine; Tumbes (45 °54′ S, 81 °05′ W) (MUSM-HEL 3310) (new geographical record) (present study) [33].

Remarks

This species was recorded as *Sprostoniella* sp. by Chero et al. [33], but a taxonomic study of the voucher specimens deposited in the MUSM-HEL showed that this record is conspecific with *S*. *lamothei* Pérez-Ponce de Léon & Mendoza-Garfias, 2000. Status: valid species.


**Order Dactylogyridea Bychowsky, 1937**



**Family Calceostomatidae Parona & Perugia, 1890**



***Calceostoma* sp.**


*Orthopristis chalceus*; gills; marine; La Libertad, Piura [16].

Remarks

No specimens in collections.


**Family Dactylogyridae Bychowsky, 1933**



***Ameloblastella edentensis* Mendoza-Franco, Mendoza-Palmero & Scholz, 2016**


*Hypophthalmus edentatus*; gills; freshwater; Loreto (03°42′ S, 73°17′ W) [34].

*Hypophthalmus edentatus*; gills; freshwater; Loreto (03°47′ S, 73°21′ W) (holotype, IPCAS M-622; paratypes and vouchers, IPCAS M-622; paratypes, USNM 1418027; USNM 1418028) [35].

Note

Sequences were deposited in GenBank under the accession number KP056255 for the partial 28S rRNA [34].

Remarks

This species was recorded as *Ameloblastella* sp. 16 by Mendoza-Palmero et al. [34]. Remarks

Status: valid species.


***Ameloblastella formatrium* Mendoza-Franco, Mendoza-Palmero & Scholz, 2016**


Pimelodidae gen. sp.; gills; freshwater; Loreto (03°47′ S, 73°21′ W) (holotype, IPCAS M-624; paratypes, IPCAS M-624; paratypes, USNM 1418030) [35].

Remarks

Status: valid species.


***Ameloblastella martinae* Mendoza-Palmero, Rossin, Irigoitia & Scholz, 2020**


*Sorubim lima* (type host), *Hemisorubim platyrhynchos*; gills; freshwater; Loreto (03°46′ S, 73°15′ W) (holotype, IPCAS M-724; paratype, IPCAS M-724; paratypes, CNHE 11260–11261; CNHE 11262) [36].

*Pseudoplatystoma punctifer*; gills; freshwater; Loreto [37].

Note

Sequences were deposited in GenBank under the accession number MT174.

Remarks

Status: valid species.

*Ameloblastella peruensis* Mendoza-Franco, Mendoza-Palmero & Scholz, 2016

*Hypophthalmus* sp.; gills; freshwater; Loreto (holotype, IPCAS M-623; paratypes, USNM 1418029) [35].

Remarks

Status: valid species.


***Ameloblastella* sp1.**


*Hassar* sp.; gills; freshwater; Loreto (03°45′ S, 73°15′ W) [34].

Note

Sequences were deposited in GenBank under the accession number KP056253 for the partial 28S rDNA [34].

Remarks

This species was recorded as *Ameloblastella* sp. by Mendoza-Palmero et al. [34].


***Ameloblastella* sp2.**


*Hypophthalmus edentatus*; gills; freshwater; Loreto (03°42′ S 73°17′ W) [34].

Note

Sequences were deposited in GenBank under the accession number KP056233 for the partial 28S rDNA.

Remarks

Referred to as *Ameloblastella* sp23. by Mendoza-Palmero et al. [34].


***Ameloblastella unapi* Mendoza-Franco & Scholz, 2009**


*Calophysus macropterus*; gills; freshwater; Loreto (holotype, USNM 1396529; paratype, USNM 1396530, paratype and vouchers, IPCAS M-482) [38].

*Calophysus macropterus*; gills; freshwater; Loreto [37].

Remarks

Status: valid species.


***Ameloblastella unapioides* Mendoza-Franco, Mendoza-Palmero & Scholz, 2016**


*Sorubim lima*; gills; freshwater; Loreto [34].

*Sorubim lima* (type host); gills; freshwater; Loreto (holotype, IPCAS M-625; paratypes and vouchers, IPCAS M-625; paratypes, USNM 1418031; USNM 1418032) [35]. 

*Pimelodus* sp.; gills; freshwater; Loreto (IPCAS M-625; USNM 1418033) [35].

Note

Sequences were deposited in GenBank under the accession number KP056254 for the partial 28S rDNA [34].

Remarks

Referred to as *Ameloblastella* sp8. by Mendoza-Palmero et al. [34]. Status: valid species.


***Amphithecium calycinum* Boeger & Kritsky, 1988**


*Pygocentrus nattereri*; gills; freshwater; Loreto (CHURP 708) [39].

*Pygocentrus nattereri*; gills; freshwater; Loreto (LAPYSA M-22) [37].

Remarks

The voucher deposited in the CHURP is not available because this collection has been deactivated. Status: valid species.


***Amphithecium camelum* Boeger & Kritsky, 1988**


*Pygocentrus nattereri*; gills; freshwater; Loreto (CHURP 716) [39].

*Pygocentrus nattereri*; gills; freshwater; Loreto (LAPYSA M-23) [37].

Remarks

The voucher deposited in the CHURP is not available because this collection has been deactivated. Status: valid species.


***Amphithecium cataloensis* Boeger & Kritsky, 1988**


*Pygocentrus nattereri*; gills; freshwater; Loreto (LAPYSA M-24) [37].

Remarks

Status: valid species.


***Amphithecium falcatum* Boeger & Kritsky, 1988**


*Pygocentrus nattereri*; gills; freshwater; Loreto (LAPYSA M-25) [37].

Remarks

Status: valid species.


***Amphithecium junki* Boeger & Kritsky, 1988**


*Pygocentrus nattereri*; gills; freshwater; Loreto (LAPYSA M-26) [37].

Remarks

Status: valid species. 


***Anacanthorus acuminatus* Kritsky, Boeger & Van Every, 1992**


*Triportheus angulatus*; gills; freshwater; Loreto (LAPYSA M-10) [40].

Remarks

Status: valid species.


***Anacanthorus amazonicus* Van Every & Kritsky, 1992**


*Pygocentrus nattereri*; gills; freshwater; Loreto (LAPYSA M-21) [37].

Remarks

Status: valid species.


***Anacanthorus camposbaci* Morey, Aliano & Grandez, 2019**


*Myloplus schomburgkii*; gills; freshwater; Loreto (03°51′ S, 73°23′ W) (holotype, MUSM-HEL 388; paratypes, MUSM-HEL 3881a-c; paratypes, INPA 796 a-f) [41].

*Myloplus schomburgkii*; gills; freshwater; Loreto [37].

Remarks

The species was originally described as *Anacanthorus camposbacae* Morey, Aliano & Grandez, 2019 by Morey et al. [41]. However, the specific name contains a malformed suffix, and the scientific name should be corrected to *A*. *camposbaci* [17]. Status: valid species.


***Anacanthorus carmenrosae* Morey, Aliano & Grandez, 2019**


*Myloplus schomburgkii*; gills; freshwater; Loreto (03°51′ S, 73°23′ W) (holotype, MUSM-HEL3882; paratypes, MUSM-HEL3882 a-c; paratypes, INPA 800 a-f) [41].

*Myloplus schomburgkii*; gills; freshwater; Loreto [37].

Remarks

Status: valid species.


***Anacanthorus chaunophallus* Kritsky, Boeger & Van Every, 1992**


*Triportheus angulatus*; gills; freshwater; Loreto (LAPYSA M-11) [40].

Remarks

Status: valid species.


***Anacanthorus chelophorus* Kritsky, Boeger & Van Every, 1992**


*Triportheus angulatus*; gills; freshwater; Loreto (LAPYSA M-12) [40]).

Remarks

Status: valid species.


***Anacanthorus cultro* Morey, Sarmiento, Chu, Cruces & Chero, 2023**


*Mylossoma albiscopum*; gills; freshwater; Loreto (4°18′ S, 74°17′ W) (holotype, MUSM-HEL 5372; paratypes, MUSM-HEL 5373 a-e; paratypes, LAPYSA M-99 a-d) [42].

Remarks

Status: valid species.


***Anacanthorus euryphallus* Kritsky, Boeger & Van Every, 1992**


*Triportheus angulatus*; gills; freshwater; Loreto (LAPYSA M-13) [40].

Remarks

Status: valid species.


***Anacanthorus femoris* Morey, Sol-Sol & Cachique, 2020**


*Brycon amazonicus*; gills; freshwater; Loreto (holotype, MUSM-HEL4429; paratypes, MUSM-HEL4428 a-f) [43].

*Brycon amazonicus*; gills; freshwater; Loreto [37].

Remarks

Status: valid species.


***Anacanthorus kukamensis* Morey, Sol-Sol & Cachique, 2020**


*Brycon amazonicus*; gills; freshwater; Loreto (holotype, MUSM-HEL4427; paratypes, MUSM-HEL4426 a-c) [43].

*Brycon amazonicus*; gills; freshwater; Loreto [37].

Remarks

Status: valid species.


***Anacanthorus lygophallus* Kritsky, Boeger & Van Every, 1992**


*Triportheus angulatus*; gills; freshwater; Loreto (LAPYSA M-14) [40].

Remarks

Status: valid species.


***Anacanthorus pedanophallus* Kritsky, Boeger & Van Every, 1992**


*Myloplus schomburgkii*; gills; freshwater; Loreto (LAPYSA M-27) [37].

Remarks

Status: valid species.


***Anacanthorus penilabiatus* Boeger, Husak & Martins, 1995**


*Piaractus brachypomus*; gills; freshwater; Loreto (LAPYSA M-28) [37].

*Pygocentrus nattereri*; gills; freshwater; Loreto (LAPYSA M-28) [37].

*Colossoma macropomum*, *Piaractus brachypomus*; gills; freshwater; Madre de Dios (UNMSM-LPFSZ 479–483) (present study).

Remarks

Status: valid species.


***Anacanthorus pithophallus* Kritsky, Boeger & Van Every, 1992**


*Triportheus angulatus*; gills; freshwater; Loreto (LAPYSA M-15) [40].

Remarks

Status: valid species.


***Anacanthorus ramosissimus* Van Every & Kritsky, 1992**


*Pygocentrus nattereri*; gills; freshwater; Loreto (LAPYSA M-29) [37].

Remarks

Status: valid species.


***Anacanthorus rarus* Morey, Sol-Sol & Cachique, 2020**


*Brycon amazonicus*; gills; freshwater; Loreto (holotype, MUSM-HEL4430; paratype, MUSM-HEL4431) [43].

*Brycon amazonicus*; gills; freshwater; Loreto [37].

Remarks

Status: valid species.


***Anacanthorus reginae* Boeger & Kritsky, 1988**


*Pygocentrus nattereri*; gills; freshwater; Loreto (CHURP 710) [39].

*Pygocentrus nattereri*; gills; freshwater; Loreto (LAPYSA M-30) [37].

Remarks

The voucher deposited in the CHURP is not available because this collection has been deactivated. Status: valid species.


***Anacanthorus sabaloi* Morey, Sol-Sol & Cachique, 2020**


*Brycon amazonicus*; gills; freshwater; Loreto (holotype, MUSM-HEL4432; paratypes, MUSM-HEL4433 a-d) [43].

*Brycon amazonicus*; gills; freshwater; Loreto [37].

Remarks

Status: valid species.


***Anacanthorus scapanus* Van Every & Kritsky, 1992**


*Pygocentrus nattereri*; gills; freshwater; Loreto (LAPYSA M-31) [37].

Remarks

Status: valid species.


***Anacanthorus* sp1.**


*Myloplus schomburgkii*; gills; freshwater; Loreto [44].

Remarks

No specimens in collections.


***Anacanthorus* sp2.**


*Colossoma macropomum*; gills; freshwater; Loreto [45].

Remarks

Referred to as Anacanthorinae gen. sp. By Vargas et al. [45]. Vargas et al. [45] did not mention where their specimens were deposited, but an analysis of the photographs they provided indicates that these specimens belong to the genus *Anacanthorus*. 


***Anacanthorus spathulatus* Kritsky, Thatcher & Kayton, 1979**


*Colossoma macropomum*; gills; freshwater; Loreto (03°49′ S, 73°19′ W) [46].

*Colossoma macropomum*; gills; freshwater; Loreto (LAPYSA M-32) [37].

*Colossoma macropomum*, *Piaractus brachypomus*; gills; freshwater; Madre de Dios (UNMSM-LPFSZ 484–486) (present study).

Remarks

Status: valid species.


***Anacanthorus spiralocirrus* Kritsky, Thatcher & Kayton, 1979**


*Brycon amazonicus*; gills; freshwater; Loreto [43].

*Brycon amazonicus*; gills; freshwater; Loreto (LAPYSA M-33) [37].

Remarks

Status: valid species.


***Anacanthorus stachophallus* Kritsky, Boeger & Van Every, 1992**


*Pygocentrus nattereri*; gills; freshwater; Loreto (CHURP 720) [39].

*Pygocentrus nattereri*; gills; freshwater; Loreto (LAPYSA M-34) [37].

Remarks

The voucher deposited in the CHURP is not available because this collection has been deactivated. Status: valid species.


***Anacanthorus thatcheri* Boeger & Kritsky, 1988**


*Pygocentrus nattereri*; gills; freshwater; Loreto (CHURP 707) [39].

*Pygocentrus nattereri*; gills; freshwater; Loreto (LAPYSA M-35) [37].

Remarks

The voucher deposited in the CHURP is not available because this collection has been deactivated. Status: valid species.


***Ancistrohaptor falciferum* Agarwal & Kritsky, 1998**


*Triportheus angulatus*; gills; freshwater; Loreto (LAPYSA M-16) [40].

Remarks

Status: valid species.


***Ancistrohaptor falcunculum* Agarwal & Kritsky, 1998**


*Triportheus angulatus*; gills; freshwater; Loreto (LAPYSA M-17) [40].

Remarks

Status: valid species.


***Apedunculata discoidea* Cuglianna, Cordeiro & Luque, 2009**


*Prochilodus nigricans*; gills; freshwater; Loreto (03°41′ S, 73°12′ W) [47].

Remarks

No specimens in collections. Status: valid species.


***Aphanoblastella aurorae* Mendoza-Palmero, Scholz, Mendoza- Franco & Kuchta, 2012**


*Goeldiella eques*; gills; freshwater; Loreto (03°47′ S, 73°21′ W) (holotype, IPCAS M-524; paratypes, IPCAS M-524; paratypes USNM 1400341–1400343; paratypes, NHMUK 2012.3.15.26-30; USNM 1400344–1400347) [48].

*Goeldiella eques*; gills; freshwater; Loreto (03°47′ S, 73°21′ W) [34].

Note

Sequences were deposited in GenBank under the accession number KP056239 for the partial 28S rDNA [34].

Remarks

Status: valid species.


***Aphanoblastella* sp.**


*Goeldiella eques*, gills; freshwater; Loreto (03°42′ S, 73°17′ W) [34]. 

Note

Sequences were deposited in GenBank under the accession numbers KP066237 and KP066238 for the partial 28S rDNA [34].

Remarks

Referred to as *Aphanoblastella* sp3. By Mendoza-Palmero et al. [34].


***Bicentenariella claudiae* Cruces, Chero, Sáez & Luque, 2021**


*Pronotogrammus multifasciatus*; gills; marine; Tumbes (3°29′ S, 80°24′ W) (holotype, MUSM-HEL 4715; paratypes, MUSM-HEL 4716a-r) [49].

Remarks

Status: valid species.


***Bicentenariella peruensis* (Cruces, Chero, Sáez & Luque, 2017) Cruces, Chero, Sáez & Luque, 2021**


*Hemanthias peruanus* (type host); gills; marine; Tumbes (3°29′ S, 80°24′ W) (holotype, MUSM-HEL 3480; paratypes, MUSM-HEL 3481–3507; paratypes, CHIOC 38852 a-c) [50].

*Hemanthias peruanus*; gills; marine; Piura (HPIA 40) [51].

*Hemanthias peruanus*; gills; marine; Tumbes (03°29′ S, 80°24′ W) [49].

Remarks

This species was described as *Parancylodiscoides peruensis* by Cruces et al. [50] and was transferred to the genus *Bicenteneriella* by Cruces et al. [49]. Status: valid species.


***Bicentenariella puertopizarroensis* Cruces, Chero, Sáez & Luque, 2021**


*Pronotogrammus multifasciatus*; gills; marine; Tumbes (03°29′ S, 80°24′ W) (holotype, MUSM-HEL 4719; paratypes, MUSM-HEL 4720a-n) [49].

Remarks

Status: valid species.


***Bicentenariella signiferi* (Cruces, Chero, Sáez & Luque, 2017) Cruces, Chero, Sáez & Luque, 2021**


*Hemanthias 11imaculat*; gills; marine; Tumbes (3°29′ S, 80°24′ W) (holotype, MUSM-HEL 3508; paratypes, MUSM-HEL 3509–3539; paratypes, CHIOC 38851a-b) [50].

*Hemanthias 11imaculat*; gills; marine; Tumbes (03°29′ S, 80°24′ W) (holotype, MUSM-3508, paratypes, MUSM-HEL3509–3539) [49].

Remarks

This species was described as *Parancylodiscoides signiferi* by Cruces et al. [50] and was transferred to the genus *Bicenteneriella* by Cruces et al. [49]. Status: valid species. 


***Bicentenariella sinuosa* Cruces, Chero, Sáez & Luque, 2021**


*Pronotogrammus multifasciatus*; gills; marine; Tumbes (03°29′ S, 80°24′ W) (holotype, MUSM-HEL 4717; paratypes, MUSM-HEL 4718a-l) [49].

Remarks

Status: valid species.


***Biotodomella mirospinata* Morey, Arimuya & Boeger, 2019**


*Biotodoma cupido*; gills; freshwater; Loreto (3°45′ S, 73°17′ W) (holotype, INPA 785a; paratypes, INPA 785 b-g; 797 a-j) [52].

*Biotodoma cupido*; gills; freshwater; Loreto (3°45′ S, 73°17′ W) (LAPYSA M-66) [53].

Remarks

Status: valid species.


***Boegeriella conica* (Mendoza-Palmero, Mendoza-Franco, Acosta & Scholz, 2019) Mendoza-Palmero & Hsiao, 2020**


*Ageneiosus vittatus*, gills; freshwater; Loreto (03°42′ S, 73°17′ W) [34].

*Platynematichthys notatus* (type host), *Brachyplatystoma juruense*; gills; freshwater; Loreto (03°42′ S, 73°17′ W) (holotype, IPCAS M-699; paratypes, IPCAS M-699; paratypes, CNHE 11161; IPCAS M-699; CNHE 11162) [54].

*Platynematichthys notatus*; gills; freshwater; Loreto (IIAP-MM-0035) [55].

Remarks

Referred to as Dactylogyridae gen sp10. By Mendoza-Palmero et al. [34]. Mendoza-Palmero et al. [54] proposed the genus *Walteriella* Mendoza-Palmero, Mendoza-Franco, Acosta & Scholz, 2019 to accommodate their new species, *W*. *conica* and *W*. *ophiocirrus*. However, the generic name had already been assigned to a genus of soldier beetles, *Walteriella* Kazantsev, 2001 (Coleoptera: Cantharidae). Thus, Mendoza-Palmero & Hsiao [56] proposed the genus *Boegeriella* Mendoza-Palmero & Hsiao, 2020 and transferred *W*. *conica* and *W*. *ophiocirrus* to *Boegeriella*. Status: valid species. 

Note

Sequences were deposited in GenBank under the accession numbers KP056225, KP056226, and KP056227 [34] and MK834513 and MK834514 [54] for the partial 28S rRNA. 


***Boegeriella ophiocirrus* (Mendoza-Palmero, Mendoza-Franco, Acosta & Scholz, 2019) Mendoza-Palmero & Hsiao, 2020**


*Platystomatichthys sturio*; gills; freshwater; Loreto (holotype, IPCAS M-700; paratypes, IPCAS M-700; paratypes, CNHE 11163; IPCAS M-700; CNHE 11164) [54].

Remarks

Mendoza-Palmero et al. [54] proposed the genus *Walteriella* Mendoza-Palmero, Mendoza-Franco, Acosta & Scholz, 2019 to accommodate their new species, *W*. *conica* and *W*. *ophiocirrus*. However, the generic name had already been assigned to a genus of soldier beetles, *Walteriella* Kazantsev, 2001 (Coleoptera: Cantharidae). Thus, Mendoza-Palmero & Hsiao [56] proposed the genus *Boegeriella* Mendoza-Palmero & Hsiao, 2020 and transferred *W*. *conica* and *W*. *ophiocirrus* to *Boegeriella*. Status: valid species. 

Note

Sequences were deposited in GenBank under the accession numbers MK834511, MK834512, and MK834515 for the partial 28S rRNA [54].


***Brotulella laurafernandae* Cruces, Chero & Luque, 2023**


*Brotula clarkae*; gills; marine; Tumbes (03°29′ S, 80°24′ W) (holotype, MUSM-HEL 5132; paratypes, MUSM-HEL 5133a-l) [57].

Remarks

Status: valid species.

Note

The sequence was deposited in GenBank under the accession number OR860318 for the partial 28S rDNA [57].


***Brotulella luisahelenae* Cruces, Chero & Luque, 2023**


*Brotula clarkae*; gills; marine; Tumbes (03°29′ S, 80°24′ W) (holotype, MUSM-HEL 5134; paratypes, MUSM-HEL 5135a-k) [57]).

Remarks

Status: valid species.

Note

The sequence was deposited in GenBank under the accession number OR860321 for the partial 28S rDNA [57].


***Cichlidogyrus sclerosus* Paperna & Thurston, 1969**


*Oreochromis niloticus*; gills; freshwater; San Martín (CPMP 503) (new geographical record) (present study).

Remarks

Status: valid species.


***Chclidogyrus* sp.**


*Oreochromis niloticus*; gills; freshwater; San Martin [58].

Remarks

No specimens in collections. 


***Cichlidogyrus tilapiae* Paperna, 1960**


*Oreochromis niloticus*; gills; freshwater; Loreto (03°48′ S, 73°19′ W) (LAPYSA M-E-2) [59].

Remarks

Status: valid species.


***Cleidodiscus* sp.**


*Pimelodella yuncensis*; adult; gills; freshwater; La Libertad [60].

Remarks

No specimens in collections.


***Cosmetocleithrum baculum* Yamada, Yamada & da Silva, 2020**


*Trachelyopterus* sp.; gills; freshwater; Loreto (03°46′ S, 73°15′ W) [61].

Remarks

No specimens in collections. Status: valid species.

Note

The sequence was deposited in GenBank under the accession number ON982893 for the partial 28S rDNA [61].

***Cosmetocleithrum bifurcum*** 
**Mendoza-Franco, Mendoza-Palmero & Scholz, 2016**

*Hassar orestis*; gills; freshwater; Loreto (03°45′ S, 73°15W) [34].

*Hassar orestis* (type host); gills; freshwater; Loreto (03°45′ S, 73°15′ W) (holotype, IPCAS M-627; paratypes and vouchers, IPCAS M-627; paratypes, USNM 1418035; USNM 1418036) [35].

Remarks

Referred to as *Cosmetocleithrum* sp8. By Mendoza-Palmero et al. [34]. Status: valid species.

Note

Sequences were deposited in GenBank under the accession numbers KP056216 and KP056217 for the partial 28S rDNA [34].


***Cosmetocleithrum bulbocirrus* Kritsky, Thatcher & Boeger, 1986**


*Pterodoras granulosus*; gills; freshwater; Loreto (03°47′ S, 73°15W) (IPCAS M-525; USNM 1400348–1400349; NHMUK 2012.3.14.37-43) [48].

Remarks

Status: valid species.


***Cosmetocleithrum confusum* Kritsky, Thatcher & Boeger, 1986**


*Oxydoras niger*; gills; freshwater; Loreto (CHURP 704) [15].

*Oxydoras niger*; gills; freshwater; Loreto (03°46′ S, 73°15′ W) [61].

Remarks

This species was reported as *Cosmetocleithrum 13imacula* Kritsky, Thatcher & Boeger, 1986. However, the specific name contains a malformed suffix, and the correct name is *C*. *confusum*. The voucher deposited in the CHURP is not available because this collection has been deactivated. Status: valid species.

Note

Sequences were deposited in GenBank under the accession number ON982791 for the partial 28S rDNA [61].


***Cosmetocleithrum falsunilatum* **
**Feronato, Razzolini, Morey & Boeger, 2022**


*Megalodoras uranoscopus*; gills; freshwater; Loreto (03°46′ S, 73°15′ W) (holotype, CHIOC 39731a; paratypes, CHIOC 39731b-m) [62].

Remarks

Status: valid species.

Note

Sequences were deposited in GenBank under the accession number OM971057 for the partial 28S rDNA [62].


***Cosmetocleithrum gigas* Morey, Cachique & Babilonia, 2019**


*Oxydoras niger*; gills; freshwater; Loreto (03°46′ S, 73°15′ W) (holotype, MUSM-HEL 3880; paratypes, MUSM-HEL 3880 a-c) [63].

*Oxydoras niger*; gills; freshwater; Loreto (03°46′ S, 73°15′ W) [61].

Remarks

Status: valid species.

Note

Sequences were deposited in GenBank under the accession number ON982794 for the partial 28S rDNA [61].


***Cosmetocleithrum gussevi* Kritsky, Thatcher & Boeger, 1986**


*Oxydoras niger*; gills; freshwater; Loreto (CHURP 702) [15].

*Oxydoras niger*; gills; freshwater; Loreto (03°46′ S, 73°15′ W) [61].

Remarks

The voucher deposited in the CHURP is not available because this collection has been deactivated. Status: valid species.

Note

Sequences were deposited in GenBank under the accession number ON982795 for the partial 28S rDNA [61].


***Cosmetocleithrum infinitum* Morey, Rojas & Cachique, 2022**


*Anadoras grypus*; gills; freshwater; Loreto (holotype, MUSM-HEL4421; paratypes, MUSM-HEL4420a-d) [64].

Remarks

Status: valid species.

***Cosmetocleithrum laciniatum*** 
**Yamada, Yamada, Silva & Anjos, 2017**

*Trachelyopterus* sp.; gills; freshwater; Loreto (03°46′ S, 73°15′ W) [61].

Remarks

Status: valid species.

Note

Sequences were deposited in GenBank under the accession number ON982796 for the partial 28S rDNA [61].


***Cosmetocleithrum parvum* Kritsky, Thatcher & Boeger, 1986**


*Oxydoras niger*; gills; freshwater; Loreto (03°46′ S, 73°15′ W) [61].

Remarks

Status: valid species.

Note

Sequences were deposited in GenBank under the accession number ON982792 for the partial 28S rDNA [61].


***Cosmetocleithrum rarum* Kritsky, Thatcher & Boeger, 1986**


*Oxydoras niger*; gills; freshwater; Loreto (03°46′ S, 73°15′ W) [61].

Remarks

Status: valid species.

Note

Sequences were deposited in GenBank under the accession number ON982797 for the partial 28S rDNA [61].


***Cosmetocleithrum sobrinus* Kritsky, Thatcher & Boeger, 1986**


*Oxydoras niger*; gills; freshwater; Loreto (CHURP 703) [15].

Remarks

The voucher deposited in the CHURP is not available because this collection has been deactivated. Status: valid species.


***Cosmetocleithrum tortum* Mendoza-Franco, Mendoza-Palmero & Scholz, 2016**


*Nemadoras hemipeltis*; gills; freshwater; Loreto (holotype, IPCAS M-626; paratype and voucher, IPCAS M-626; paratype, USNM 1418034) [35].

Remarks

Status: valid species.


**Dactylogyridae gen. sp1.**


*Colossoma macropomum*; gills; freshwater; Loreto [65].

Remarks

No specimens in collections.


**Dactylogyridae gen. sp2.**


*Ageneiosus vittatus*; gills; freshwater; Loreto (03°42′ S, 73°17′ W) [34].

Note

Sequences were deposited in GenBank under the accession number KP056218 for the partial 28S rDNA [34].

Remarks

Referred to as Dactylogiridae gen. sp4. By Mendoza-Palmero et al. [34].


**Dactylogyridae gen. sp3.**


*Platynematichthys notatus*; gills; freshwater; Loreto (03°47′ S, 73°21′ W) [34]. 

Note

Sequences were deposited in GenBank under the accession numbers KP056220–KP056224 for the partial 28S rDNA [34].

Remarks

Referred to as Dactylogiridae gen. sp9. By Mendoza-Palmero et al. [34].


**Dactylogyridae gen. sp4.**


*Sorubim lima*; gills; freshwater; Loreto (03°46′ S, 73°15′ W) [34]. 

Note

Sequences were deposited in GenBank under the accession number KP056228 for the partial 28S rDNA [34].

Remarks

Referred to as Dactylogiridae gen. sp12. By Mendoza-Palmero et al. [34].


**Dactylogyridae gen. sp5.**


*Hypophthalmus 15imaculat*; gills; freshwater; Loreto (3°42′ S, 73°17′ W) [34]. 

Note

Sequences were deposited in GenBank under the accession numbers KP056229–KP056230 for the partial 28S rDNA [34].

Remarks

Referred to as Dactylogiridae gen. sp13. By Mendoza-Palmero et al. [34].


**Dactylogyridae gen. sp6.**


*Hypophthalmus 15imaculat*; gills; freshwater; Loreto (03°47′ S, 73°21′ W) [34]. 

Note

Sequences were deposited in GenBank under the accession number KP056231 for the partial 28S rDNA [34].

Remarks

Referred to as Dactylogiridae gen. sp18. By Mendoza-Palmero et al. [34].


**Dactylogyridae gen. sp7.**


*Hypophthalmus 15imaculat*; gills; freshwater; Loreto (03°47′ S, 73°21′ W) [34]. 

Note

Sequences were deposited in GenBank under the accession number KP056232 for the partial 28S rDNA [34]. 

Remarks

Referred to as Dactylogiridae gen. sp23. By Mendoza-Palmero et al. [34].


**Dactylogyridae gen. sp8.**


*Platynematichthys notatus*; gills; freshwater; Loreto (03°47′ S, 73°21′ W) [34]. 

Note

Sequences were deposited in GenBank under the accession number KP056234 for the partial 28S rDNA [34]. 

Remarks

Referred to as Dactylogiridae gen. sp26. By Mendoza-Palmero et al. [34].


**Dactylogyridae gen. sp9.**


*Colossoma macropomum*; gills; freshwater; Loreto [66].

Remarks

No specimens in the collection. 


**Dactylogyrus sp.**


*Carassius auratus*; gills, epidermis; freshwater; Lima [67].

Apistogramma sp., Cichlasoma sp., Pterophyllum scalare, Symphysodon aequifasciatus; gills; freshwater; Peruvian Amazon [68].

*Colossoma macropomum*; gills; freshwater; Ucayali [69].

Remarks

No specimens in collections.


***Dactylogyrus vastator* Nybelin, 1924**


*Cyprinus carpio*; gills; freshwater; La Libertad [70].

Remarks

No specimens in collections. Status: valid species.


***Dawestrema cycloancistrioides* Kritsky, Boeger & Thatcher, 1985**


*Arapaima gigas*; gills; freshwater; Loreto (3°49′ S, 73°19′ W) [71].

*Arapaima gigas*; gills; freshwater; Loreto (3°49′ S, 73°19′ W) [72].

*Arapaima gigas*; gills; freshwater; Loreto [73].

Remarks

No specimens in collections. Status: valid species.


**Dawestrema cycloancistrium Price & Nowlin, 1967**


*Arapaima gigas*; gills; freshwater; Loreto (CHURP 701) [15].

*Arapaima gigas*; gills; freshwater; Loreto [74].

*Arapaima gigas*; gills; freshwater; Loreto [75].

*Arapaima gigas*; gills; freshwater, Loreto (3°49′ S, 73°19′ W) [71].

*Arapaima gigas*; gills; freshwater, Loreto [76].

*Arapaima gigas*; gills; freshwater; Loreto [73].

*Arapaima gigas*; gills; freshwater, Loreto [77].

*Arapaima gigas*; gills; freshwater, Loreto (05°54′ S, 76°05′ W) (CHIOC 38.656 a-c, 38657 a-c) [78].

*Arapaima gigas*; gills, freshwater; Loreto [79].

*Arapaima gigas;* gills, freshwater; Loreto (LAPYSA M-36) [80].

Remarks

The voucher deposited in the CHURP is not available because this collection has been deactivated. Status: valid species.


***Demidospermus brevicirrus* Mendoza-Palermo, Scholz, Mendoza-Franco & Kuchta, 2012**


*Pimelodus* sp.; gills; freshwater; Loreto (03°47′ S, 73°21′ W) (holotype, IPCAS M-523; paratypes and vouchers, IPCAS M-523; paratypes, USNM 1400335–1400338; vouchers, USNM 1400339, 1400340, 1400350; paratypes, NHMUK 2012.3.12.12–2012.3.12.18; voucher, NHMUK 2012.3.15.19–2012.3.15.25) [48].

Remarks

Status: valid species.


***Demidospermus centromochli* Mendoza-Franco & Scholz, 2009**


*Centromochlus heckelii*; gills; freshwater; Loreto (holotype and paratype on a single slide in USNM 1396527; paratype, USNM 1396528; paratype, IPCAS M-483) [38].

Remarks

Status: valid species.


***Demidospermus curvovaginatus* Mendoza-Palmero & Scholz, 2011**


*Pimelodus* sp.; gills; freshwater; Loreto (holotype, IPCAS M-513; paratype, USNM 1399527; voucher, USNM 1399528–1399529; paratype, BMNH 2011.3.31.5; voucher, BMNH 2011.3.31.6–2011.3.31.9) [81].

Remarks

Status: valid species.


***Demidospermus doncellae* Morey, Rojas, Dávila, Chu & de Pina, 2023**


*Pseudoplatystoma punctifer*; gills; freshwater; Loreto (holotype, MUSM-HEL 4233; paratype, MUSM-HEL 4234a-c) [82].

Remarks

Status: valid species.


***Demidospermus macropteri* Mendoza-Franco & Scholz, 2009**


*Calophysus macropterus*; gills; freshwater; Loreto (holotype, USNM 1396531; paratypes, USNM 1396532) [38].

*Calophysus macropterus*; gills; freshwater; Loreto [37].

Remarks

Status: valid species.


***Demidospermus mortenthaleri* Mendoza-Palermo, Scholz, Mendoza-Franco & Kuchta, 2012**


*Brachyplatystoma juruense*; gills; freshwater; Loreto (03°47′ S, 73°21′ W) (holotype IPCAS M-522; paratypes and vouchers, IPCAS M-522; paratypes USNM 1400329–1400331; paratypes, NHMUK 2012.3.15.1-5; USNM 1400332–1400334; NHMUK 2012.3.15.6-11.) [48].

*Brachyplatystoma juruense*; gills; freshwater; Loreto (03°47′ S, 73°21′ W) [34].

Remarks

Status: valid species.

Note

Sequences were deposited in GenBank under the accession numbers KP056245 and KP056246 for the partial 28S rDNA [34].


***Demidospermus peruvianus* Mendoza-Palmero & Scholz, 2011**


*Pimelodus ornatus* (type host), *P*. *blochii*; gills; freshwater; Loreto (holotype, IPCAS M-515; paratypes, USNM 1399524, vouchers; USNM 1399525–1399526; paratypes, BMNH 2011.3.31.1–2011.3.31.2; vouchers, BMNH 2011.3.31.3–2011.3.31.4) [81].

*Pimelodus* sp.; gills; freshwater; Loreto (holotype, IPCAS M-513; paratype, USNM 1399527; voucher, USNM 1399528–1399529; paratype, BMNH 2011.3.31.5; voucher, BMNH 2011.3.31.6–2011.3.31.9) [81].

Remarks

Status: valid species.


***Demidospermus* sp1.**


*Brachyplatystoma vaillantii*; gills; freshwater; Loreto (03°42′ S, 73°17′ W) [34].

Note

Sequences were deposited in GenBank under the accession number KP056235 for the partial 28S rDNA [34].

Remarks

Referred to as *Demidospermus* sp11. By Mendoza-Palmero et al. [34].


***Demidospermus* sp2.**


*Brachyplatystoma vaillantii*; gills; freshwater; Loreto (03°42′ S, 73°17′ W) [34].

Note

Sequences were deposited in GenBank under the accession number KP056236 for the partial 28S rDNA [34].

Remarks

Referred to as *Demidospermus* sp23. By Mendoza-Palmero et al. [34].


***Demidospermus* sp3.**


*Calophysus macropterus*; gills; freshwater; Loreto (LAPYSA M-37) [37].


***Demidospermus striatus* Mendoza-Palmero & Scholz, 2011**


*Pimelodus* sp., *P*. *blochii*; gills; freshwater; Loreto (holotype, IPCAS M-514; paratype, USNM 1399530; voucher, USNM 1399531; paratypes, BMNH 2011.3.31.10–2011.3.31.11; vouchers, BMNH 2011.3.31.12–2011.3.31.13) [81].

Remarks

Status: valid species.


***Demidospermus wilberi* Cruces, Santillán, Silvera & Chero, 2023**


*Loricaria* sp.; gills; freshwater; Madre de Dios (12°35′ S, 69°10′ W) (holotype, MUSM-HEL 5364; paratype, MUSM-HEL MUSM-HEL 5365a-i) [83].

Remarks

Status: valid species.


***Enallothecium aegidatum* (Boeger & Kritsky, 1988) Kritsky, Boeger & Jégu, 1998**


*Pygocentrus nattereri*; gills; freshwater; Loreto (LAPYSA M-38) [37].

Remarks

Status: valid species.


***Euryhaliotrema chaoi* Kritsky & Boeger, 2002**


*Plagioscion squamosissimus*; gills; freshwater; Loreto (03°45′ S, 73°17′ W) (USNM 1386542, 1386544, 1386545) [84].

Remarks

Status: valid species.


**Euryhaliotrema lovejoyi Kritsky & Boeger, 2002**


*Plagioscion squamosissimus*; gills; freshwater; Loreto (03°45′ S, 73°17′ W) (USNM 1386548) [84].

Remarks

Status: valid species.


***Euryhaliotrema luisae* Cruces, Chero & Luque, 2018**


*Calamus brachysomus*; gills; marine; Tumbes (3°29′ S, 80°24′ W) (holotype, MUSM-HEL 3782; paratypes, MUSM-HEL 3783a-g; paratypes, CHIOC 39076a-f.) [85].

Remarks

Status: valid species.


***Euryhaliotrema magnopharyngis* Cruces, Chero & Luque, 2018**


*Calamus brachysomus*; gills; marine; Tumbes (03°29′ S, 80°24′ W) (holotype, MUSM-HEL 3784; paratypes, MUSM-HEL 3785a-c; paratypes, CHIOC 39077a-b) [85].

Remarks

Status: valid species. 


***Euryhaliotrema monacanthus* Kritsky & Boeger, 2002**


*Plagioscion squamosissimus*; gills; freshwater; Loreto (03°45′ S, 73°17′ W) (USNM 1386581) [84].

*Plagioscion squamosissimus*; gills; freshwater; Loreto (04°02′ S, 73°09′ W) (USNM 1386582) [84].

Remarks

Status: valid species.


***Euryhaliotrema paralonchuri* (Luque & Iannocone, 1989) Kritsky & Boeger, 2002**


*Paralonchurus peruanus*; gills; marine; Lima (12°30′ S, 76°50’) (holotype, CHURP 511, paratype, CHURP 512) [86].

*Paralonchurus peruanus*; gills; marine; Ancash, Lima [87].

*Paralonchurus peruanus*; gills; marine; Lima (12°30′ S, 76°50’) [88].

*Paralonchurus peruanus*; gills; marine; Lima [89].

*Paralonchurus peruanus*; gills; marine; Lima (12°30′ S, 76°50’) (USNM 1386595) [84].

*Paralonchurus peruanus*; gills; marine; Lima (12°30′ S, 76°50’) (unpublished data, USNM 1376672).

Remarks

This species was described as a member of the genus *Pseudohaliotrema* Yamaguti, 1953 by Luque and Iannocone [86] and was transferred to *Euryhaliotrema* Kritsky & Boeger, 2002 by Kritsky and Boeger [84]. According to Kritsky and Boeger [84], this species needs to be redescribed. The type material (holotypes and paratypes) deposited in the CHURP is not available because this collection has been deactivated and all material has been lost. Status: valid species.


**Euryhaliotrema potamocetes Kritsky & Boeger, 2002**


*Plagioscion squamosissimus*; gills; freshwater; Loreto (03°45′ S, 73°17′ W) (USNM 1386557–1386560) [84].

Remarks

Status: valid species.


***Euryhaliotrema sagmatum* Kritsky & Boeger, 2002**


*Umbrina xanti*; gills; marine; Tumbes (03°29′ S 80°24′ W) (MUSM-HEL 4723) [22].

Remarks

Status: valid species.


***Euryhaliotrema succedaneus* Kritsky & Boeger, 2002**


*Plagioscion squamosissimus*; gills; freshwater; Loreto (03°45′ S, 73°17′ W) (holotype, USNM 1386584; paratype, USNM 1386585–1386589; paratype, HWML 16428; paratype, MNHN 25HG Tg 155–156) [84].

Remarks

Status: valid species.


**Euryhaliotrema thatcheri Kritsky & Boeger, 2002**


*Plagioscion squamosissimus*; gills; freshwater; Loreto (03°45′ S, 73°17′ W) (USNM 1386566–1386569, 1386572) [84].

*Plagioscion squamosissimus*; gills; freshwater; Loreto (04°02′ S, 73°09′ W) (USNM 1386573) [84].

Remarks

Status: valid species.


***Gonocleithrum aruanae* Kritsky, Thatcher & Boeger, 1983**


*Osteoglossum bicirrhosum*; gills; freshwater; Loreto (05°40′ S, 74°19′ W) [90].

Remarks

No specimens in collections. Status: valid species.


***Gonocleithrum coenoideum* Kritsky, Thatcher & Boeger, 1983**


*Osteoglossum bicirrhosum*; gills; freshwater; Loreto (05°04′ S, 74°19′ W) [90].

Remarks

No specimens in collections. Status: valid species.


***Gonocleithrum cursitans* Kritsky, Thatcher & Boeger, 1983**


*Osteoglossum bicirrhosum*; gills; freshwater; Loreto [91].

*Osteoglossum bicirrhosum*; gills; freshwater; Loreto (05°04′ S, 74°19′ W) [90].

Remarks

No specimens in collections. Status: valid species.


***Gussevia alii* (Molnár, Hanek & Fernando, 1974) Kritsky, Thatcher & Boeger, 1986**


*Cichlasoma amazonarum*; gills; freshwater; Loreto (03°44′ S, 73°17′ W) (LAPYSA M-77) [53].

Remarks

Status: valid species.


***Gussevia alioides* Kritsky, Thatcher, and Boeger, 1986**


*Heros severus*; gills; freshwater; Loreto (USNM 1397200) [92].

Remarks

This species was redescribed by Mendoza-Franco et al. [92]. Status: valid species. 


***Gussevia arilla* Kritsky, Thatcher & Boeger, 1986**


*Cichla monoculus*; gills; freshwater; Amazon basing [93].

Note

Sequences were deposited in GenBank under the accession numbers ON368040 for the partial 18S rRNA and ITS1 and ON261196 for the partial 28S rRNA.

Remarks

No specimens in collections. Status: valid species.


***Gussevia asota* Kritsky, Thatcher & Boeger, 1989**


*Astronotus ocellatus*; gills; freshwater; Loreto (USNM 1397199, IPCAS M-498) [92].

*Astronotus ocellatus*; gills; freshwater; Loreto (03°49′ S, 73°19′ W) [94].

*Astronotus ocellatus*; gills; freshwater; Amazon basing [95].

*Astronotus ocellatus;* gills; freshwater; Loreto [37].

*Astronotus ocellatus*; gills; freshwater; Loreto (03°44′ S, 73°17′ W) (LAPYSA M-65) [53].

Note

Sequences were deposited in GenBank under the accession numbers ON368041 for the partial 18S rRNA and ITS1 and ON261214 for the partial 28S rRNA [93].

Remarks

This species was redescribed by Mendoza-Franco et al. [92]. Status: valid species.


***Gussevia astronoti* Kritsky, Thatcher & Boeger, 1989**


*Astronotus ocellatus*; gills; freshwater; Amazon basing [93].

*Astronotus ocellatus;* gills; freshwater; Loreto (LAPYSA M-39) [37].

*Astronotus ocellatus;* gills; freshwater; Loreto (LAPYSA M-39) [53].

Remarks

Status: valid species.

Note

Sequences were deposited in GenBank under the accession numbers ON368042 for the partial 18S rRNA and ITS1 and ON261215 for the partial 28S rRNA [93].


***Gussevia cichlasomatis* (Molnár, Hanek & Fernando, 1974) Kritsky, Thatcher & Boeger, 1986**


*Aequidens tetramerus*; gills; freshwater; Loreto (03°55′ S, 73°22′ W) (LAPYSA M-94) [53].

*Cichlasoma amazonarum*; gills; freshwater; Loreto (03°44′ S, 73°17′ W) (LAPYSA M-76) [53].

Remarks

Status: valid species.


***Gussevia dispar* Kritsky, Thatcher & Boeger, 1986**


*Heros efasciatus*; gills; freshwater; Loreto (03°44′ S, 73°17′ W) (LAPYSA M-83) [53].

Remarks

Status: valid species. 


***Gussevia disparoides* Kritsky, Thatcher & Boeger, 1986**


*Cichlasoma amazonarum*, *Heros severus*; gills; freshwater; Loreto (USNM 1397201–1397202) [92].

*Cichlasoma amazonarum*; gills; freshwater; Amazon basing [93].

*Heros efasciatus*; gills; freshwater; Loreto (03°44′ S, 73°17′ W) (LAPYSA M-84) [53]. 

Note

Sequences were deposited in GenBank under the accession numbers ON368044 for the partial 18S rRNA and ITS1 and ON261197 for the partial 28S rRNA [93].

Remarks

This species was redescribed by Mendoza-Franco et al. [92]. Status: valid species.


***Gussevia longihaptor* (Mizelle & Kritsky, 1969) Kritsky, Thatcher & Boeger, 1986**


*Cichla monoculus*; gills; freshwater; Loreto (IPCAS M-497, USNM 1397197) [92].

*Cichla monoculus*; gills; freshwater; Loreto [95].

Remarks

This species was redescribed by Mendoza-Franco et al. [92]. Status: valid species.


***Gussevia rogersi* Kritsky, Thatcher & Boeger, 1989**


*Astronotus ocellatus*; gills; freshwater; Loreto (LAPYSA M-40) [37].

*Astronotus ocellatus;* gills; freshwater; Loreto (LAPYSA M-40) [53].

Remarks

Status: valid species.


**Gussevia spiralocirra Kohn & Paperna, 1964**


*Pterophyllum scalare*; gills; freshwater; Loreto (INPA PA272–1–PA272–3; USNM 1374193; HWML 22955) [96].

*Pterophyllum scalare*; gills; freshwater; Amazon basing [93].

*Pterophyllum scalare*; gills; freshwater; Loreto (3°44′ S, 73°17′ W) (LAPYSA M-67) [53].

Note

Sequences were deposited in GenBank under the accession numbers ON368049 for the partial 18S rRNA and ITS1 and ON261203 for the partial 28S rRNA [93].

Remarks

This species was redescribed by Kritsky et al. [96]. Status: valid species.


***Gussevia* sp1.**


*Acaronia nassa*; gills; freshwater; Loreto (3°44′ S, 73°17′ W) (LAPYSA M-91) [53].


***Gussevia* sp2.**


*Acaronia nassa*; gills; freshwater; Loreto (3°44′ S, 73°17′ W) (LAPYSA M-92) [53].


***Gussevia* sp3.**


*Acaronia nassa*; gills; freshwater; Loreto (3°44′ S, 73°17′ W) (LAPYSA M-93) [53].


***Gussevia* sp4.**


*Aequidens tetramerus*; gills; freshwater; Loreto (03°55′ S, 73°22′ W) (LAPYSA M-95) [53].


***Gussevia* sp5.**


*Aequidens tetramerus*; gills; freshwater; Loreto (03°55′ S, 73°22′ W) (LAPYSA M-96) [53].


***Gussevia* sp6.**


*Aequidens tetramerus*; gills; freshwater; Loreto (03°55′ S, 73°22′ W) (LAPYSA M-97) [53].


***Gussevia* sp7.**


*Cichlasoma amazonarum*; gills; freshwater; Loreto (03°44′ S, 73°17′ W) (LAPYSA M-78) [53].


***Gussevia* sp8.**


*Cichlasoma amazonarum*; gills; freshwater; Loreto (03°44′ S, 73°17′ W) (LAPYSA M-79) [53].


***Gussevia* sp9.**


*Heros efasciatus*; gills; freshwater; Loreto (03°44′ S, 73°17′ W) (LAPYSA M-85) [53].


***Gussevia* sp10.**


*Cichlasoma amazonarum*; gills; freshwater; Amazon basing [93].

Note

Sequences were deposited in GenBank under the accession numbers ON368043 for the partial 18S rRNA and ITS1 and ON261198 for the partial 28S rRNA [93].

Remarks

Referred to as *Gussevia* sp. 1 by Seidlová et al. [93]. 


***Gussevia* sp11.**


*Cichlasoma amazonarum*; gills; freshwater; Amazon basing [93].

Note

Sequences were deposited in GenBank under the accession number ON281754 for the partial 28S rRNA.

Remarks

Referred to as *Gussevia* sp. 3 by Seidlová et al. [93].


***Gussevia* sp12.**


*Mesonauta 21imacula*; gills; freshwater; Amazon basing [93].

Note

Sequences were deposited in GenBank under the accession numbers ON368047 for the partial 18S rRNA and ITS1 and ON261201 for the partial 28S rRNA [93].

Remarks

Referred to as *Gussevia* sp. 5 by Seidlová et al. [93].


***Gussevia tucunarensis* Kritsky, Thatcher & Boeger, 1986**


*Chaetobranchus semifasciatus*; gills; freshwater, Loreto (3°49′ S, 73°19′ W) [97].

*Cichla monoculus*; gills; freshwater; Amazon basing [93].

*Cichla monoculus*; gills; freshwater; Loreto (3°44′ S, 73°17′ W) (LAPYSA M-74) [53].

Remarks

Status: valid species.

Note

Sequences were deposited in GenBank under the accession numbers ON368050 for the partial 18S rRNA and ITS1 and ON261204 for the partial 28S rRNA [93].


***Gussevia undulata* Kritsky, Thatcher & Boeger, 1986**


*Cichla monoculus*; gills; freshwater; Loreto (USNM 1397198) [92].

*Cichla monoculus*; gills; freshwater; Loreto (3°49′ S, 73°19′ W) [98].

*Cichla monoculus*; gills; freshwater; Loreto [95].

*Cichla monoculus*; gills; freshwater; Amazon basing [93].

*Cichla monoculus;* gills; freshwater; Loreto [37].

Note

Sequences were deposited in GenBank under the accession numbers ON368051 for the partial 18S rRNA and ITS1 and ON261205 for the partial 28S rRNA [93].

Remarks

This species was redescribed by Mendoza-Franco et al. [92]. Status: valid species.


***Haliotrema diplotaenia* Cruces, Chero & Luque, 2018**


*Bodianus diplotaenia*; gills; marine; Tumbes (45°54′ S, 81°05′ W) (holotype, MUSM-HEL 3786; paratypes, MUSM-HEL 3787a-g; paratypes, CHIOC 39078a-g) [85].

Remarks

Status: valid species.


***Haliotrema saezae* Cruces, Chero & Luque, 2018**


*Bodianus diplotaenia*; gills; marine; Tumbes (3°29′ S, 80°24′ W) (holotype, MUSM-HEL 3788; paratypes, MUSM-HEL 3789a-c; paratypes, CHIOC 39079a-c) [85].

Remarks

Status: valid species.


***Haliotrema sanchezae* Cruces, Chero, Sáez & Luque, 2017**


*Scarus perrico*; gills; marine; Tumbes (3°29′ S, 80°24′ W) (holotype, MUSM-HEL 3471; paratypes, MUSM-HEL 3472–3479; paratypes, CHIOC 38853a-c) [50].

Remarks

Status: valid species.


***Haliotrematoides mediohamides* Kritsky & Mendoza-Franco in Kritsky, Yang & Sun, 2009**


*Calamus brachysomus*; gills; marine; Tumbes (03°29’ S, 80°24’ W) (MUSM-HEL 4850a-i) [99].

Remarks

This species was redescribed by Cruces et al. [99]. Status: valid species. 


***Haliotrematoides prolixohamus* Kritsky & Mendoza-Franco in Kritsky, Yang & Sun, 2009**


*Calamus brachysomus*; gills; marine; Tumbes (3°29’ S, 80°24’ W) (CPMP 504) (new geographical record) (present study).

Remarks

Status: valid species.


***Hamatopeduncularia* sp.**


*Galeichthys peruvianus*; gills; marine; Lima (12°30’ S, 76°50’ W) (USNM 1376673) [100].


***Heteropriapulus heterotylus* (Jogunoori, Kritsky & Venkatanarasaiah, 2004) Kritsky, 2007**


*Pterygoplichthys pardalis*; gills; freshwater; Loreto (LAPYSA M-41) [37].

Remarks

Status: valid species.


***Jainus amazonensis* Kritsky, Thatcher & Kayton, 1980**


*Brycon cephalus*; gills; freshwater; Loreto (3°49′ S, 73°19′ W) [101].

Remarks

Status: valid species.


***Jainus peruensis* Cruces, Santillán, Silvera, Morey & Chero, 2024**


*Brycon amazonicus*; gills; freshwater; Madre de Dios (12°35′ S, 69°10′ W) (holotype, MUSM-HEL 5392; paratype, MUSM-HEL MUSM-HEL 5393a-i) [102].

Remarks

Status: valid species.


***Jainus* sp1.**


*Triportheus angulatus*; gills; freshwater; Loreto (LAPYSA M-18) [40].


***Jainus* sp2.**


*Triportheus angulatus*; gills; freshwater; Loreto (LAPYSA M-19) [40].


***Jainus* sp3.**


*Triportheus angulatus*; gills; freshwater; Loreto (LAPYSA M-20) [40].


***Ligophorus mugilinus* (Hargis, 1955) Euzet & Suriano, 1977**


*Mugil cephalus*; gills; freshwater; La Libertad [70].

Remarks

This species was reported as *Haliotrema mugilinus* Hargis, 1955 by Jara and Escalante [70]. Status: valid species.


***Mexicana iannaconi* Chero, Cruces, Sáez & Alvariño, 2014**


*Haemulon steindachneri*; gills; marine; Lima (12°09′ S 76°56′ W) (holotype, CPMP 112; paratype, CPMP 113) [103].

Remarks

Status: valid species.


***Mexicana* sp.**


*Anisotremus scapularis*; gills; marine; Lima (12°18′ S, 76°53′ W) [104].

*Anisotremus scapularis*; gills; marine; Lima (CPMP-UNFV 074-075) [26]. 


***Mymarothecium boegeri* Cohen & Kohn, 2005**


*Colossoma macropomum*; gills; freshwater; Loreto (LAPYSA M-48) [37].

Remarks

Status: valid species.


***Mymarothecium iiapensis* Morey, Aliano & Grandez, 2019**


*Colossoma macropomum*; gills; freshwater; Loreto (03°49′ S, 73°20′ W) (holotype, MUSM-HEL 3884; paratypes, MUSM-HEL 3883a-b; paratypes, INPA 798 a-d) [41].

*Colossoma macropomum*; gills; freshwater; Loreto [37].

Remarks

Status: valid species.


***Mymarothecium galeolum* Kritsky, Boeger & Jegu, 1996**


*Pygocentrus nattereri*; gills; freshwater; Loreto (LAPYSA M-49) [37].

Remarks

Status: valid species.


***Mymarothecium* sp.**


*Colossoma macropomum*; gills; freshwater; Loreto [45].

Remarks

Referred to as Ancyrocephalinae gen. sp. By Vargas et al. [45]. 


***Mymarothecium viatorum* Boeger, Piasecki & Sobecka, 2002**


*Piaractus brachypomus*; gills; freshwater; Loreto [105].

*Piaractus brachypomus;* gills; freshwater; Loreto (LAPYSA M-47a) [37].

*Pygocentrus nattereri;* gills; freshwater; Loreto (LAPYSA M-47b) [37].

*Colossoma macropomum*; gills; freshwater; Madre de Dios (UNMSM-LPFSZ 471–474) (present study).

Remarks

Status: valid species. 


***Mymarothecium tantaliani* Cayulla-Quispe, Mondragón-Martínez, Rojas-De Los Santos, Garcia-Candela, Babilonia-Medina & Martínez-Rojas, 2020**


*Colossoma macropomum*; gills; freshwater; Madre de Dios (holotype, MUSM-HEL 4662; paratype, MUSM-HEL 4663) [106].

Remarks

Status: valid species.


***Nanayella aculeatrium* Acosta, Mendoza-Palmero, da Silva & Scholz, 2019**


*Sorubim lima*; gills; freshwater, Loreto (03°46′ S, 73°15′ W) [34].

*Sorubim lima*; gills; freshwater; Loreto (03°46′ S, 73°19′ W) (holotype, IPCAS M-694; paratypes, IPCAS M-694; hologenophore, IPCAS M-694) [107].

Note

The sequence was deposited in GenBank under the accession number KP056228 for the partial 28S rDNA [34].

Remarks

Referred to as Dactylogyridae gen. sp12. By Mendoza-Palmero et al. [34]. Status: valid species.

***Nanayella megorchis*** 
**(Mizelle & Kritsky, 1969) Acosta, Mendoza-Palmero, da Silva & Scholz, 2019**

*Sorubim lima*; gills; freshwater; Loreto (IPCAS M-698; hologenophore, IPCAS M-698) [107].

Remarks

This species was described as *Urocleidoides megorchis* by Mizelle and Kritsky, 1969 and was transferred to the genus *Nanayella* by Acosta et al. [107]. Status: valid species. 

Note

Sequences were deposited in GenBank under the accession numbers MK367405, MK367406, and MK367407 for the partial 28S rDNA [107].


***Nanayella* sp.**


*Pseudoplatystoma punctifer*; gills; freshwater; Loreto (LAPYSA M-50) [37].

Remarks

Status: valid species.


***Notothecioides llewellyni* Kritsky, Boeger & Jegu, 1997**


*Pygocentrus nattereri*; gills; freshwater; Loreto (LAPYSA M-51) [37].

Remarks

Status: valid species.


***Notozothecium agusti* Cruces, Santillán, Silvera & Chero, 2023**


*Brycon amazonicus*; gills; freshwater; Madre de Dios (12°35′ S, 69°10′ W) (holotype, MUSM-HEL 5366; paratypes, MUSM-HEL 5367a-i) [83].

Remarks

Status: valid species.


***Notozothecium bethae* Kritsky, Boeger & Jégu, 1996**


*Myloplus schomburgkii*; gills; freshwater; Loreto (3°49′ S, 73°19′ W) [108].

*Myloplus schomburgkii;* gills; freshwater; Loreto (LAPYSA M-52) [37].

Remarks

Status: valid species.


***Notozothecium janauachensis* Belmont-Jegu, Domingues, & Martins, 2004**


*Colossoma macropomum*; gills; freshwater; Loreto (LAPYSA M-53) [37].

*Colossoma macropomum*; gills; freshwater; Madre de Dios (UNMSM-LPFSZ 475–478) (present study).

Remarks

Status: valid species.


***Notozothecium minor* Boeger & Kritsky, 1988**


*Pygocentrus nattereri*; gills; freshwater; Loreto (LAPYSA M-54) [37].

Remarks

Status: valid species.


***Notothecium mizellei* Boeger & Kritsky, 1988**


*Pygocentrus nattereri*; gills; freshwater; Loreto (CHURP 715) [39].

Remarks

The voucher deposited in the CHURP is not available because this collection has been deactivated. Status: valid species.


***Notozothecium nanayense* Morey, Aliano & Grandez, 2019**


*Myloplus schomburgkii*; gills; freshwater; Loreto (03°51′ S, 73°23′ W) (holotype, MUSM-HEL 3883; paratypes, MUSM-HEL 3883a-c; paratypes, INPA 799 a-f) [41].

*Myloplus schomburgkii*; gills; freshwater; Loreto [37].

Remarks

The species was originally described as *Notozothecium nanayensis* Morey, Aliano & Grandez, 2019 by Morey et al. [41]. However, the specific name contains a malformed suffix, and the scientific name should be corrected to *N*. *nanayense* [17]. Status: valid species.


***Notozothecium palometae* Morey, Sarmiento, Chu, Cruces & Chero, 2023**


*Mylossoma albiscopum*; gills; freshwater; Loreto (4°18′ S, 74°17′ W) (holotype, MUSM-HEL 5374; paratypes, MUSM-HEL 5375a-e; paratypes, LAPYSA M-100 a-d) [42].

Remarks

Status: valid species. 


**Notozothecium penetrarum Boeger & Kritsky, 1988**


*Pygocentrus nattereri*; gills; freshwater; Loreto (CHURP 709) [39].

Remarks

The voucher deposited in the CHURP is not available because this collection has been deactivated. Status: valid species.


***Notozothecium* sp1.**


*Pygocentrus nattereri*; gills; freshwater; Loreto (LAPYSA M-55) [37].


***Notozothecium* sp2.**


*Myloplus schomburgkii*; gills; freshwater; Loreto [44].


***Onchocleidus* sp.**


*Lebiasina 25imaculate*; gills; freshwater; La Libertad [109].

Remarks

Reported as *Oncocleidus* sp. By Jara and Escalante [109].


**Parancylodiscoides chaetodipteri Caballero & Bravo-Hollis, 1961**


*Parapsettus panamensis*; gills; marine; Tumbes (MUSM-HEL 3247; CPMP-UNFV 172a-c) [110].

*Parapsettus panamensis*; gills; marine; Tumbes (45°54′ S, 81°05′ W) (MUSM-HEL 3247) [33].

Remarks

Status: valid species.


***Peruanella aureagarciae* (Morey, Rojas, Dávila, Chu & de Pina, 2023) Cruces, Santillán, Silvera, Murrieta & Chero, 2024**


*Pseudoplatystoma punctifer*; gills; freshwater; Loreto (holotype, MUSM-HEL 4231; paratype, MUSM-HEL 4232a-c) [102].

Remarks

This species was described as *Demidospermus aureagarciae* Morey, Rojas, Dávila, Chu & de Pina, 2023 by Morey et al. [82] and was transferred to the genus *Peruanella* by Cruces et al. [102]. Status: valid species.

***Peruanella madredediosensis*** 
**Cruces, Santillán, Silvera, Murrieta & Chero, 2024**

*Brachyplatystoma tigrinum*; gills; freshwater; Madre de Dios (12°35′ S, 69°10′ W) (holotype, MUSM-HEL 5390; paratype, MUSM-HEL MUSM-HEL 5391a-i) [102].

Remarks

Status: valid species.


***Philocorydoras alcantarai* **
**Morey, Rojas & Panduro, 2022**


*Corydoras ambiacus*; gills; freshwater; Loreto (03°27′ S, 72°48′ W) (holotype, MUSM-HEL 4227; paratypes, MUSM-HEL 4228a-e) [111].

Remarks

Status: valid species.


***Philocorydoras beleniensis* Morey, Rojas & Panduro, 2022**


*Corydoras ambiacus*; gills; freshwater; Loreto (03°27′ S, 72°48′ W) (holotype, MUSM-HEL 4225; paratypes, MUSM-HEL 4226a-e) [111].

Remarks

Status: valid species.


***Philocorydoras jumboi* Morey, 2021**


*Corydoras multiradiatus*; gills; freshwater; Loreto (03°27′ S, 72°48′ W) (holotype, MUSM-HEL 4223; paratypes, MUSM-HEL 4224a-e) [112].

Remarks

Status: valid species.

***Philocorydoras maltai*** 
**Morey, Rojas & Panduro, 2022**

*Corydoras splendens*; gills; freshwater; Loreto (03°27′ S, 72°48′ W) (holotype, MUSM-HEL 4221; paratypes, MUSM-HEL 4222a-e) [111].

Remarks

Status: valid species.


**Philocorydoras multiradiatus Morey, 2021**


*Corydoras multiradiatus*; gills; freshwater; Loreto (03°27′ S, 72°48′ W) (holotype, MUSM-HEL 4219; paratypes, MUSM-HEL 4220a-e) [112].

Remarks

Status: valid species.


***Philocorydoras peruensis* Morey, 2021**


*Corydoras splendens*; gills; freshwater; Loreto (03°27′ S, 72°48′ W) (holotype, MUSM-HEL 4229; paratypes, MUSM-HEL 4230a-e) [112].

Remarks

Status: valid species.


***Pronotogrammella boegeri* Cruces, Chero, Sáez & Luque, 2020**


*Pronotogrammus multifasciatus*; gills; marine; Tumbes (03°29′ S, 80°24′ W) (holotype, MUSM-HEL 4430; paratypes, MUSM-HEL 431a-j; paratypes, CHIOC 39218a-e) [113].

Remarks

Status: valid species.


***Pronotogrammella multifasciatus* Cruces, Chero, Sáez & Luque, 2020**


*Pronotogrammus multifasciatus*; gills; marine; Tumbes (3°29′ S, 80°24′ W) (holotype, MUSM-HEL 4434; paratypes, MUSM-HEL 4435a-b; paratype, CHIOC 39219) [113].

Remarks

Status: valid species.


***Pronotogrammella scholzi* Cruces, Chero, Sáez & Luque, 2020**


*Pronotogrammus multifasciatus*; gills; marine; Tumbes (3°29′ S, 80°24′ W) (holotype, MUSM-HEL 4432; paratypes, MUSM-HEL 4433a-f; paratypes, CHIOC 39220a-e) [113].

*Hemanthias peruanus*; gills; marine; Piura (HPIA 42) [51].

Remarks

Status: valid species.


***Rhinonastes pseudocapsaloideum* Kritsky, Thatcher & Boeger, 1988**


*Prochilodus nigricans*; gills and nasal cavity; freshwater; Loreto [114].

Remarks

Status: valid species.


***Rhinoxenus piranhus* Kritsky, Boeger & Thatcher, 1988**


*Pygocentrus nattereri*; gills; freshwater; Amazonas (CHURP 714) [39].

*Pygocentrus nattereri*; gills; freshwater; Loreto (LAPYSA M-56) [37].

Remarks

The voucher deposited in the CHURP is not available because this collection has been deactivated. Status: valid species.


***Sciadicleithrum edgari* Paschoal, Scholz, Tavares-Dias & Luque, 2016**


*Chaetobranchus flavescens*; gills; freshwater; Loreto (LAPYSA M-57) [37].

*Chaetobranchus flavescens*; gills; freshwater; Loreto (03°45′ S, 73°17′ W) (LAPYSA M-73) [53].

Remarks

Status: valid species.


***Sciadicleithrum ergensi* Kritsky, Thatcher & Boeger, 1989**


*Cichla monoculus*; gills; freshwater; Loreto [95].

*Cichla monoculus*; gills; freshwater; Amazon basing [93].

*Cichla monoculus*; gills; freshwater; Loreto (LAPYSA M-58) [37].

Remarks

Status: valid species.

Note

Sequences were deposited in GenBank under the accession numbers ON368053, ON368052 for the partial 18S rRNA and ITS1, and ON261216, ON261218 for the partial 28S rRNA [93].


***Sciadicleithrum iphthimum* Kritsky, Thatcher & Boeger, 1989**


*Pterophyllum scalare*; gills; freshwater; Loreto (03°44′ S, 73°17′ W) (LAPYSA M-68) [53].

Remarks

Status: valid species.


***Sciadicleithrum satanopercae* Yamada, Takemoto, Bellay y Pavanelli, 2009**


*Satanoperca jurupari*; gills; freshwater; Loreto (USNM 1397203, IPCAS M-499) [92].

*Satanoperca jurupari*; gills; freshwater; Loreto (03°45′ S, 73°17′ W) (LAPYSA M-69) [53].

Remarks

This species was redescribed by Mendoza-Franco et al. [92]. Status: valid species.


***Sciadicleithrum* sp1.**


*Aequidens tetramerus*; gills; freshwater; Loreto (03°55′ S, 73°22′ W) (LAPYSA M-97) [53].


***Sciadicleithrum* sp2.**


*Biotodoma cupido*; gills; freshwater; Loreto (03°45′ S, 73°17′ W) (LAPYSA M-88) [53].


***Sciadicleithrum* sp3.**


*Bujurquina peregrinabunda*; gills; freshwater; Loreto (03°55′ S, 73°25′ W) (LAPYSA M-89) [53].


***Sciadicleithrum* sp4.**


*Bujurquina peregrinabunda*; gills; freshwater; Loreto (03°55′ S, 73°25′ W) (LAPYSA M-90) [53].


***Sciadicleithrum* sp5.**


*Crenicichla johanna*; gills; freshwater; Loreto (03°44′ S, 73°17′ W) (LAPYSA M-81) [53].


***Sciadicleithrum* sp6.**


*Crenicichla johanna*; gills; freshwater; Loreto (03°44′ S, 73°17′ W) (LAPYSA M-82) [53].


***Sciadicleithrum* sp7.**


*Crenicichla johanna*; gills; freshwater; Loreto (03°55′ S, 73°22′ W) (LAPYSA M-86) [53].


***Sciadicleithrum* sp8.**


*Mesonauta mirificus*; gills; freshwater; Loreto (03°55′ S, 73°22′ W) (LAPYSA M-87) [53].


***Sciadicleithrum* sp9.**


*Mesonauta festivus*; gills; freshwater; Amazon basing [93].

Note

Sequences were deposited in GenBank under the accession numbers ON368059 for the partial 18S rRNA and ITS1 and ON261209 for the partial 28S rRNA [93].

Remarks

Referred to as *Sciadicleithrum* sp. 4 by Seidlová et al. [93].


***Sciadicleithrum umbilicum* Kritsky, Thatcher & Boeger, 1989**


*Cichla monoculus*; gills; freshwater; Amazon basing [93].

*Cichla monoculus*; gills; freshwater; Loreto (03°44′ S, 73°17′ W) (LAPYSA M-75) [53].

Remarks

Status: valid species.

Note

Sequences were deposited in GenBank under the accession numbers ON368065 for the partial 18S rRNA and ITS1 and ON261217 for the partial 28S rRNA [93].


***Sciadicleithrum uncinatum* Kritsky, Thatcher & Boeger, 1989**


*Cichla monoculus*; gills; freshwater; Amazon basing [93].

Remarks

No specimens in collections. Status: valid species.


***Sciadicleithrum variabile* (Mizelle & Kritsky, 1969) Kritsky, Thatcher & Boeger, 1989**


*Cichlasoma amazonarum*; gills; freshwater; Loreto (IPCAS M-280) [92].

*Cichlasoma amazonarum*; gills; freshwater; Amazon basing [93].

*Cichlasoma amazonarum*; gills; freshwater; Loreto (03°44′ S, 73°17′ W) (LAPYSA M-72) [53].

*Heros efasciatus*; gills; freshwater; Loreto (03°44′ S, 73°17′ W) (LAPYSA M-71) [53].

*Symphysodon tarzoo*; gills; freshwater; Loreto (03°47′ S, 73°21′ W) (LAPYSA M-70) [53].

*Cichlasoma amazonarum*; gills; freshwater; Loreto (unpublished data, USNM 1397204).

Note

Sequences were deposited in GenBank under the accession numbers ON368038 for the partial 18S rRNA and ITS1 and ON261194 for the partial 28S rRNA [93].

Remarks

The species was reported as *Sciadicleithrum variabilum* (Mizelle & Kritsky 1969) Kritsky, Thatcher & Boeger, 1989 by Mendoza-Franco et al. [92]. The specific epithet represents a malformed suffix, and the correct name is *Sciadicleithrum variabile* [17]. Referred to as *Sciadicleithrum variabilum* (Mizelli & Kritsky, 1969) form A by Seidlová et al. [93]. This species was redescribed by Mendoza-Franco et al. [92]. Status: valid species.


***Tereancistrum curimba* Lizama, Takemoto & Pavanelli, 2004**


*Prochilodus nigricans*; gills; freshwater; Loreto (03°41′ S, 73°12′ W) [47].

*Prochilodus nigricans*; gills; freshwater; Loreto (LAPYSA M-59) [37].

Remarks

Status: valid species.


***Tereancistrum toksonum* Lizama, Takemoto & Pavanelli, 2004**


*Prochilodus nigricans*; gills; freshwater; Loreto (3°41′ S, 73°12′ W) [47].

Remarks

No specimens in collections. Status: valid species.


***Trianchoratus acleithrium* Price & Berry, 1966**


*Trichopodus trichopterus*; gills; freshwater; Loreto (03°45′ S, 73°17′ W) (MUSM-3885a-d) [115].

Remarks

Status: valid species.


***Trinigyrus peregrinus* Nitta & Nagasawa, 2016**


*Pterygoplichthys pardalis*; gills; freshwater: Loreto (LAPYSA M-60) [37].

Remarks

Status: valid species.


***Trinidactylus cichlasomatis* Hanek, Molnár & Fernando, 1974**


*Cichlasoma amazonarum*; gills; freshwater; Loreto (03°44′ S, 73°17′ W) (LAPYSA M-80) [53].

Remarks

Status: valid species.


***Tucunarella cichlae* **
**Mendoza-Franco, Scholz & Rozkosná, 2010**


*Cichla monoculus*; gills; freshwater; Loreto (20°52′ N, 90°09′ W) (holotype, USNM 1397195; paratype, USNM 1397196; paratype, IPCAS M-496) [92].

Remarks

Status: valid species.


***Tylosuricola amatoi* Iannacone & Luque, 1990**


*Strongylura scapularis*; gills; marine; Lima (12°30′ S, 76°50′ W) (holotype, USNM 1376667; paratype, USNM 1376668; paratype; CHURP) [14].

Remarks

The paratype deposited in the CHURP is not available because this collection has been deactivated. Status: valid species.


***Unibarra paranoplatensis* Suriano & Incorvaia, 1995**


*Aguarunichthys torosus*; gills; freshwater; Loreto (03°47′ S, 73°21′ W) [34].

Remarks

Status: valid species.

Note

Sequences were deposited in GenBank under the accession number KP056219 for the partial 28S rDNA.


***Unilatus brittani* Mizelle, Kritsky & Crane, 1968**


*Pterygoplichthys anisitsi*; gills; freshwater; Loreto (03°46′ S, 73°15′ W) (IPCAS M-529; USNM 1400356; NHMUK 2012.3.15.53-54) [48].

*Pterygoplichthys pardalis*; gills; freshwater; Loreto (LAPYSA M-61) [37].

Remarks

Status: valid species.


***Unilatus unilatus* Mizelle & Kritsky, 1967**


*Pterygoplichthys anisitsi*; gills; freshwater; Loreto (03°46′ S, 73°15′ W) (IPCAS M-530) [48].

*Pterygoplichthys pardalis*; gills; freshwater; Loreto (LAPYSA M-62) [37].

Remarks

Status: valid species.


***Urocleidoides eremitus* Kritsky, Thatcher & Boeger, 1986**


*Pygocentrus nattereri*; gills; freshwater; Amazonas (CHURP 711) [39].

Remarks

The voucher deposited in the CHURP is not available because this collection has been deactivated. Status: valid species.


***Urocleidus* sp.**


*Andinoacara rivulatus*; gills; freshwater; La Libertad [109].


***Vancleaveus cicinnus* Kritsky, Thatcher & Boeger, 1986**


*Pseudoplatystoma punctifer*; gills; freshwater; Loreto (LAPYSA M-63) [37].

Remarks

Status: valid species.


***Vancleaveus fungulus* Kritsky, Thatcher & Boeger, 1986**


*Pseudoplatystoma fasciatum*; gills; freshwater; Loreto (3°46′ S, 73°20′ W) (IPCAS M-526; USNM 1400352) [48].

*Pseudoplatystoma punctifer*; gills; freshwater; Loreto (LAPYSA M-64) [37].

Remarks

Status: valid species.


***Vancleaveus janauacaensis* Kritsky, Thatcher & Boeger, 1986**


*Pterodoras granulosus*; gills; freshwater; Loreto (3°47′ S, 73°15′ W) (IPCAS M-527; USNM 1400353, 1400354; NHMUK 2012.3.15.44–51) [48].

*Pterodoras granulosus*; gills; freshwater; Loreto (3°47′ S, 73°15′ W) [34].

Remarks

Status: valid species.

Note

Sequences were deposited in GenBank under the accession numbers KP056240, KP056247, and KP056248 by Mendoza-Palmero et al. [34].


***Vancleaveus platyrhynchi* Kritsky, Thatcher & Boeger, 1986**


*Hemisorubim platyrhynchos*; gills; freshwater; Loreto (3°45′ S, 73°14′ W) (IPCAS M-528; USNM 1400355; NHMUK 2012.3.15.52) [48].

Remarks

Status: valid species.


**Family Diplectanidae Monticelli, 1903**



***Cynoscionella sanmarci* Chero, Cruces, Saéz & Luque, 2022**


*Cynoscion phoxocephalus*; gills; marine; Tumbes (holotype, MUSM-HEL 4907; paratypes, MUSM-HEL 4908a-d; paratypes, CHIOC 39757 a-e) [116].

Remarks

Status: valid species.


**Diplectanidae gen. sp1.**


*Cheilotrema fasciatum*; gills; marine; Lima [89].

Remarks

This unidentified species likely corresponds to the genus *Rhamnocercus* Monaco, Wood & Mizelle, 1954, which is a group of diplectanids commonly found infecting marine sciaenid fish [116]. Chero et al. [116] described *Rhamnocercus fasciatus* Chero, Cruces, Sáez & Luque, 2022, a species found on *Cheilotrema fasciatum* in Peru. This raises the possibility that Diplectanidae gen. sp1. of Oliva and Luque [90] may be conspecific with *R*. *fasciatus*. However, there is no information available regarding the scientific collection where the voucher specimens were deposited, which would be necessary to confirm this hypothesis.


**Diplectanidae gen. sp2.**


*Cheilotrema fasciatum*; gills; marine; Lima [89].

Remarks

This unidentified species likely corresponds to the genus *Rhamnocercus* Monaco, Wood & Mizelle, 1954, which is a group of diplectanids commonly found infecting marine sciaenid fish [116]. Chero et al. [116] described *Rhamnocercus fasciatus* Chero, Cruces, Sáez & Luque, 2022, a species found on *Cheilotrema fasciatum* in Peru. This raises the possibility that Diplectanidae gen. sp2. of Oliva and Luque [90] may be conspecific with *R*. *fasciatus*. However, there is no information available regarding the scientific collection where the voucher specimens were deposited, which would be necessary to confirm this hypothesis.


**Diplectanidae gen. sp3.**


*Paralonchurus peruanus*; gills; marine; Lima [89].

Remarks

This unidentified species likely corresponds to the genus *Rhamnocercus* Monaco, Wood & Mizelle, 1954, which is a group of diplectanids commonly found infecting marine sciaenid fish [116]. Chero et al. [116] described *Rhamnocercus dominguesi* Chero, Cruces, Sáez, Iannacone & Luque, 2017, a species found on *P. peruanus* in Peru. This raises the possibility that Diplectanidae gen. sp3. of Oliva and Luque [90] may be conspecific with *R. dominguesi*. However, there is no information available regarding the scientific collection where the voucher specimens were deposited, which would be necessary to confirm this hypothesis.


***Diplectanum decorum* Kritsky & Thatcher, 1984**


*Plagioscion squamosissimus*; gills; freshwater; Loreto (CHURP 717) [39].

Remarks

The voucher deposited in the CHURP is not available because this collection has been deactivated.


***Diplectanum pescadae* Kritsky & Thatcher, 1984**


*Plagioscion squamosissimus*; gills; freshwater; Loreto (CHURP 712) [39].

Remarks

The voucher deposited in the CHURP is not available because this collection has been deactivated.


***Diplectanum pisciniarius* Kritsky & Thatcher, 1984**


*Plagioscion squamosissimus*; gills; freshwater; Loreto (CHURP 713) [39].

Remarks

The voucher deposited in the CHURP is not available because this collection has been deactivated.


***Diplectanum* sp. 1.**


*Cynoscion analis*; gills; marine; Lima (12°30′ S, 76°50′ W) [117].

*Cynoscion analis*; gills; marine; Lima (12°30′ S, 76°50′ W) (MUS-HEL 1735) [118].

*Cynoscion analis*; gills; marine; Lima (CPMP 173a-b) [119].


***Diplectanum* sp. 2.**


*Cheilotrema fasciatum*; gills; marine; Lima (12°30′ S, 76°50′ W) [88].

Remarks

No specimens in collections.


***Diplectanum* sp. 3.**


*Cheilotrema fasciatum*; gills; marine; Lima (12°30′ S, 76°50′ W) [88].

Remarks

No specimens in collections.


***Diplectanum* sp. 4.**


*Paralonchurus peruanus*; gills; marine; Lima [87].

*Paralonchurus peruanus*; gills; marine; Lima (12°30′ S, 76°50′ W) [88].

Remarks

No specimens in collections.


***Pseudorhabdosynochus anulus* Mendoza-Franco, Violante-Gonzalez & Rojas-Herrera, 2011**


*Epinephelus labriformis*; gills; marine; Tumbes (03°29′ S, 80°24′ W) (MUSM-HEL 4724) [22].

Remarks

Status: valid species.


***Pseudorhabdosynochus jeanloui* Knoff, Cohen, Cárdenas, Cárdenas-Callirgos & Gomes, 2015**


*Paranthias colonus*; gills; marine; Lima (holotype, CHIOC 38016; paratypes, CHIOC 38016b-i; paratypes, MNHN HEL 540–541) [120].

Remarks

Status: valid species.


***Rhamnocercoides lambayequensis* Chero, Celso, Sáez & Luque, 2021**


*Menticirrhus elongatus*; gills; marine; Lambayeque (06°52′ S, 79°55′ W) (holotype, MUSM-HEL 4713; paratypes, MUSM-HEL 4714a-q) [121].

Remarks

Status: valid species.


***Rhamnocercoides menticirrhi* Luque & Iannacone, 1991**


*Menticirrhus ophicephalus*; gills; marine; Lima (12°30′ S, 76°50′ W) (holotype, CHURP 540; paratypes, CHURP 541–542) [122].

*Menticirrhus ophicephalus*; gills; marine; Lima (12°30′ S, 76°50′ W) [123].

*Menticirrhus ophicephalus*; gills; marine; Lima (12°30′ S, 76°50′ W) [88].

*Menticirrhus ophicephalus*; gills; marine; Lima (12°30′ S, 76°50′ W) [89].

*Menticirrhus ophicephalus*; gills; marine; Lima (12°4′ S, 77°10′ W) (neotype, MUSM-HEL3288; paraneotypes, MUSM-HEL3289–3291; paraneotypes CHIOC 38665a-d) [124].

Remarks

The type material of *Rhamnocercoides menticirrhi* Luque and Iannacone, 1991 was deposited in the CHURP. Currently, this is a deactivated collection, and all the deposited material has been lost. However, a voucher specimen of *R*. *menticirrhi* was deposited in the MUSM-HEL, but it is in poor condition. Chero et al. [124] redescribed this species based on newly collected specimens and designated neotype and paraneotypes for *R*. *menticirrhi*. Status: valid species.


***Rhamnocercus chacllae* Chero, Cruces, Sáez & Luque, 2022**


*Pareques lanfeari*; gills; marine; Lambayeque (06°52′ S, 79°55′ W) (holotype, MUSM-HEL 4873; paratypes, MUSM-HEL 4874a-h; paratype, CHIOC 39751) [125].

Remarks

Status: valid species.


***Rhamnocercus chaskae* Chero, Cruces, Sáez & Luque, 2022**


*Pareques lanfeari*; gills; marine; Lambayeque (06°52′ S, 79°55′ W) (holotype, MUSM-HEL 4875; paratypes, MUSM-HEL 4876a-i; paratypes, CHIOC 39752a-b) [125].

Remarks

Status: valid species.


***Rhamnocercus dominguesi* Chero, Cruces, Sáez, Iannacone & Luque, 2017**


*Paralonchurus peruanus*; gills; marine; Lima (12°4′ S, 77°10′ W) (holotype, MUSM-HEL 3292; paratypes, MUSM-HEL 3293–3297; paratypes, CHIOC 38666a-h) [124].

Remarks

Status: valid species.


***Rhamnocercus fasciatus* Chero, Cruces, Sáez & Luque, 2022**


*Cheilotrema fasciatum*; gills; marine; Lambayeque (06°52′ S, 79°55′ W) (holotype, MUSM-HEL 4877; paratypes, MUSM-HEL 4878a-h; paratype, CHIOC 39753) [125].

Remarks

Status: valid species.


***Rhamnocercus oliveri* Luque & Iannacone, 1991**


*Stellifer minor*; gills; marine; Lima (12°30′ S, 76°50′ W) (holotype, CHURP 543; paratypes, CHURP 544–545) [122].

*Stellifer minor*; gills; marine; Lima (12°30′ S, 76°50′ W) (MUSM-HEL1727) [126].

Remarks

According to Domingues and Boeger [127] and Chero et al. [125] *Rhamnocercus oliveri* Luque & Iannacone, 1991 needs to be redescribed. The type-specimens of this species have disappeared, and the only specimens available in the MUSM-HEL (voucher specimens) are in very poor condition. Thus, new type specimens (neotype and paraneotypes) need to be designated. Status: valid species.


***Rhamnocercus rimaci* Chero, Cruces, Sáez & Luque, 2022**


*Stellifer minor*; gills; marine; Lambayeque (06°52′ S, 79°55′ W) (holotype, MUSM-HEL4881; paratypes, MUSM-HEL 4882a-e; paratype, CHIOC 39755) [125].

Remarks

Status: valid species.


***Rhamnocercus* sp.**


*Stellifer minor*; gills; marine; La Libertad [31].

*Stellifer minor*; gills; marine; Lima (12°30′ S, 76°50′ W) [128].

*Stellifer minor*; gills; marine; Lima (12°30′ S, 76°50′ W) [129].

*Stellifer minor*; gills; marine; Lima (12°30′ S, 76°50′ W) [129].

*Stellifer minor*; gills; marine; Lima (12°30′ S, 76°50′ W) [88].

*Stellifer minor*; gills; marine; Lima (12° S) [89].

Remarks

No specimens in collections.


***Rhamnocercus stelliferi* Luque & Iannacone, 1991**


*Stellifer minor*; gills; marine; Lima (12°30′ S, 76°50′ W) (holotype, CHURP 546; paratypes, CHURP 547–548) [122].

*Stellifer minor*; gills; marine; Lima (12°30′ S, 76°50′ W) (MUSM-HEL 1728) [126].

Remarks

According to Domingues and Boeger [127] and Chero et al. [125] this species needs to be redescribed. The type-specimens of *Rhamnocercus stelliferi* Luque & Iannacone, 1991 have disappeared, and the only specimens available in the MUSM-HEL (voucher specimens) are in very poor condition. Thus, new type specimens (neotype and paraneotypes) need to be designated. Status: valid species.


***Rhamnocercus tantaleani* Chero, Cruces, Sáez & Luque, 2022**


*Stellifer minor* (type host); gills; marine; Lambayeque (06°52′ S, 79°55′ W) (holotype, MUSM-HEL4883; paratypes, MUSM-HEL 4884a-d; paratype, CHIOC 39756) [125].

Remarks

Status: valid species.


**Order Gyrodactylidea Bychowsky, 1937**



**Family Gyrodactylidae Van Beneden & Hesse, 1863**



***Accessorius peruensis* Jara, An & Cone, 1991**


*Lebiasina bimaculata*; skin; freshwater; La libertad (holotype, USNM 1376751; paratype, USNM 1376752) [130].

Remarks

Status: valid species.


***Anacanthocotyle* sp.**


*Eretmobrycon peruanus*; gills; freshwater; La Libertad [131].

Remarks

No specimens in collections.


**Gyrodactylidae gen. sp.**


*Colossoma macropomum*; gills; freshwater; Loreto [65].

Remarks

No specimens in collections.


***Gyrodactylus bimaculatus* An, Jara & Cone, 1991**


*Lebiasina bimaculata*; body washings; freshwater; La Libertad (78°72′ W, 8°21′ S) (holotype, USNM 1376756; paratype, USNM 8 1444) [132].

Remarks

Status: valid species.


***Gyrodactylus lebiasinus* An, Jara & Cone, 1991**


*Lebiasina bimaculata*; body washings; freshwater; La Libertad (78°72′ W, 8°21′ S) (holotype, USNM 1376758; paratype, USNM 1376759) [132].

Remarks

Status: valid species.


***Gyrodactylus pimelodellus* An, Jara & Cone 1991**


*Pimelodella yuncensis*; body washings; freshwater; La Libertad (79°21′ W, 7°46′ S) (holotype, USNM 1376753; paratype, USNM 1376754) [132].

Remarks

Status: valid species.


***Gyrodactylus slendrus* An, Jara & Cone, 1991**


*Lebiasina bimaculata*; body washings; freshwater; La Libertad (78°72′ W, 8°21′ S) (holotype, USNM 1376755; paratype, USNM 1376757) [132].

Remarks

Status: valid species.


***Gyrodactylus* sp1.**


*Lebiasina bimaculata*; gills; freshwater; La Libertad [131].

Remarks

No specimens in collections.


***Gyrodactylus* sp2.**


*Carassius auratus*; gills; freshwater; Lima [67].

Remarks

No specimens in collections.


***Gyrodactylus* sp3.**


*Oreochromis* spp.; gills; freshwater; Lima [133].

Remarks

No specimens in collections.


***Gyrodactylus* sp4.**


*Trichomycterus punctulatus*; adult; gills; freshwater; La Libertad [60].

Remarks

No specimens in collections.


***Gyrodactylus turnbulli* Harris, 1986**


*Poecilia reticulata*; body washings; freshwater; La Libertad (79°21′ W, 7°46′ S) (USNM 1376760) [132].

Remarks

Status: valid species.


***Oogyrodactylus farlowellae* Harris, 1983**


*Farlowella amazonum*; gills; freshwater; Amazonas (holotype, BMNH 1982.3.30.1; paratypes, BM(NH) 1982.3.30.2-9; paratype, USNM 1372593) [134].

Remarks

Status: valid species.


***Scleroductus yuncensi* Jara & Cone, 1989**


*Pimelodella yuncensis*; body washings; freshwater; La Libertad (07°46’ S, 79°21′ W) (holotype, USNM 1375971; paratypes, USNM 1375972) [135].

Remarks

Status: valid species.


**Order Monocotylidea Lebedev, 1988**



**Family Monocotylidae Taschenberg, 1879**



**Subfamily Dasybatotreminae Bychowsky, 1957**



***Anoplocotyloides chorrillensis* Luque & Iannacone, 1991**


*Pseudobatos planiceps*; gills; marine; Lima (12°30′ S, 76°50′ W) (holotype, USNM 1376670, paratype, USNM 1376671; paratype, CHURP 533) [23].

*Pseudobatos planiceps*; gills; marine; Lima (12°18′ S, 76°53′ W) (MUSM-HEL 2931) [136].

Remarks

The paratype deposited in the CHURP is not available because this collection has been deactivated. Status: valid species.


***Anoplocotyloides papillatus* (Doran, 1953) Young, 1967**


*Pseudobatos planiceps*; gills; marine; Lima (12°30′ S, 76°50′ W) (CHURP 532) [23].

*Pseudobatos planiceps*; gills; marine; Lima (12°18′ S, 76°53′ W) (MUSM-HEL 2932) [136].

Remarks

The voucher deposited in the CHURP is not available because this collection has been deactivated. Status: valid species.


***Peruanocotyle chisholmae* Chero, Cruces, Sáez & Luque, 2018**


*Hypanus dipterurus*; gills; marine; Lima (12°04′ S, 77°10′ W) (holotype, CHIOC 39080a; paratypes, CHIOC 39080b-d) [137].

Remarks

Status: valid species.


**Subfamily Heterocotylinae Chisholm, Wheeler & Beverley- Burton, 1995**



***Heterocotyle margaritae* hero, Cruces, Sáez, Santos & Luque, 2020**


*Hypanus dipterurus*; gills; marine; Lima (12°09′ S, 77°01′ W) (holotype, MUSM-HEL 4439; paratypes, MUSM-HEL 4437a-i; paratypes, CHIOS 39264a-d) [138].

*Hypanus dipterurus*; gills; marine; Tumbes (03°29′ S, 80°24′ W) (MUSM-HEL 4725a-c) [22].

Remarks

Status: valid species.


***Heterocotyle* sp.**


*Pseudobatos planiceps*; gills; marine; La Libertad [139].

Remarks

No specimens in collections.


***Potamotrygonocotyle chisholmae* Domingues & Marques, 2007**


*Potamotrygon motoro*; gills; freshwater; Madre de Dios (12°29′ S, 70°35′ W) [140].

*Potamotrygon motoro*; gills; freshwater; Loreto (03°43′ S, 73°12′ W) [140].

Remarks

Status: valid species.


***Potamotrygonocotyle dromedarius* Domingues & Marques, 2007**


*Potamotrygon motoro*; gills; freshwater; Madre de Dios (12°29′ S, 24°31′ W) [140].

Remarks

Status: valid species.


***Potamotrygonocotyle rionegrense* Domingues & Marques, 2007**


*Potamotrygon* sp.; gills; freshwater; Loreto (3°43′ S, 73°12′ W) [140].

Remarks

Status: valid species.


***Potamotrygonocotyle tsalickisi* Mayes, Brooks & Thorson, 1981**


*Potamotrygon* sp.; gills; freshwater; Loreto (3°43′ S, 73°12′ W) [140].

Remarks

Status: valid species.


**Subfamily Loimoinae Price, 1936**



***Loimopapillosum pascuali* Chero, Cruces, Sáez, Oliveira, Santos & Luque, 2021**


*Hypanus dipterurus*; gills; marine; Tumbes (3°29′ S, 80°24′ W) (holotype, MUSM-HEL 4660; paratypes, MUSM-HEL 4661a-x) [141].

Remarks

Status: valid species.

Note

Sequences were deposited in GenBank under the accession numbers MZ367711 and MZ367712 for the partial 18S and MZ367713 and MZ367714 for the partial 28S.


***Loimos scoliodoni* (Manter, 1938) Manter & Schmitt, 1944**


*Mustelus dorsalis*; gills; marine; La Libertad, Piura (IMT N° 563, UNMSM-LPFSZ-063) [142].

*Mustelus dorsalis*; gills; marine; La Libertad, Piura [16].

Remarks

The voucher that was initially deposited in the IMT has been redeposited in the UNMSM-LPFSZ, as the IMT collection has been deactivated. Referred to as *Loimos* sp. by Tantaleán et al. [142]. Status: valid species.


***Loimos* sp.**


*Stellifer minor*; gills; marine; La Libertad [31].

Remarks

No specimens in collections. *Stellifer minor* is an unusual host for any *Loimos* species.


**Subfamily Monocotylinae Taschenberg, 1879**



***Monocotyle luquei* Chero, Cruces, Iannacone, Sanchez, Minaya, Sáez & Alvariño, 2016**


*Hypanus dipterurus*; gills; marine; Lima (12°30′ S, 76°50′ W) (holotype, MUSM-HEL 3246; paratypes, MUSM-HEL 3246; CPMP 160–161) [143].

*Hypanus dipterurus*; gills; marine; Tumbes (03°29′ S, 80°24′ W) (MUSM-HEL 4726a-c) [22].

*Hypanus dipterurus*; gills; marine; Lima (12°09′ S, 77°01′ W) (MUSM-HEL 4726a-c) [22].

Remarks

Status: valid species.


**Subclass Polyopisthocotylea Van Beneden, 1858**



**Order Chimaericolidea Bychowsky, 1957**



**Family Chimaericolidae Brinkmann, 1942**



***Callorhynchicola branchialis* Brinkmann, 1952**


*Callorhinchus callorynchus*; gills; marine; Lima (12°02′ S, 77°01′ W) (UNMSM-LPFSZ 450) [32].

*Callorhinchus callorynchus*; gills; marine; Ica (13°44′ S, 76°13′ W) (HPIA 179) [144].

Remarks

Status: valid species.


**Order Diclybothriidea Bychowsky, 1957**



**Family Hexabothriidae Price, 1942**



**Callorhynchocotyle callorhynchi (Manter, 1955) Boeger, Kritsky & Pereira, 1989**


*Callorhinchus callorynchus*; gills; marine; Lima (12°09′ S, 77°01′ W) (MUSM-HEL 4727a-i) [22].

*Callorhinchus callorynchus*; gills; marine; Ica (13°44′ S, 76°13′ W) (HPIA 178) [144].

Remarks

Status: valid species.


***Callorhynchocotyle marplatensis* Suriano & Incorvaia, 1982**


*Callorhinchus callorynchus*; gills; marine; Lima (12°30′ S, 76°50′ W) (CHURP 536) [23].

*Callorhinchus callorynchus*; gills; marine; Lima [16].

Remarks

The voucher deposited in the CHURP is not available because this collection has been deactivated. Status: valid species.


***Erpocotyle* sp.**


*Mustelus dorsalis*.; gills; marine; La Libertad, Lima, Piura (UNMSM-LPFSZ-47) [16].

*Triakis maculata*; gills; marine; Lima [145].


***Hypanocotyle bullardi* Chero, Cruces, Sáez, Camargo, Santos & Luque, 2018**


*Hypanus dipterurus*; gills; marine; Lima (12°5′ S, 78°11′ W) (holotype, MUSM-HEL 3650; paratypes, MUSM-HEL 3651a-r) [146].

Remarks

Status: valid species.

Note

Sequences were deposited in GenBank under the accession numbers MG591251 for the partial 18S and MG591249 and MG591250 for the partial 28S.


***Rhinobatonchocotyle pacifica* Oliva & Luque, 1995**


*Pseudobatos planiceps*; gills; marine; Lima (12°30′ S, 76°50′ W) (paratype, USNM 1377340) [147].

*Pseudobatos planiceps*; gills; marine; Lima [148].

*Pseudobatos planiceps*; gills; marine; Lima (12°19′ S, 78°55′ W) (MUSM-HEL 2933) [136].

*Pseudobatos planiceps*; gills; marine; Lima (12°09′ S, 77°01′ W) (MUSM-HEL 4160a-e; CHIOC 40091a-c) [149].

Remarks

Tantaleán et al. [148] reported *Rhinobatonchocotyle cyclovaginatus* infecting *Pseudobatos planiceps* in Peru. However, Chero et al. [149] showed that this species was misidentified and corresponded to *R*. *pacificus*. *Rhinobatonchocotyle pacifica* was redescribed by Chero et al. [149]. Status: valid species.

Note

Sequences were deposited in GenBank under the accession numbers MH724313 for the partial 18S and MH714464 for the partial 28S [149].


**Order Mazocraeidea Bykhovsky, 1957**



**Family Allopyragraphoridae Yamaguti, 1953**



***Allopyragraphorus caballeroi* (Zerecero, 1960) Yamaguti, 1963**


*Caranx hippos*; gills; marine; Ancash (IMT 560, UNMSM-LPFSZ-60) [142].

*Caranx hippos*; gills; marine; Ancash [16].

Remarks

Reported as *Pyragraphorus caballeroi* Zerecero, 1960 by Tantaleán and Huiza [16]. The voucher that was initially deposited in the IMT has been redeposited in the UNMSM-LPFSZ, as the IMT collection has been deactivated. Status: valid species.


**Family Axinidae Monticelli, 1903**



***Axine ibanezi* Tantaleán, 1975**


*Exocoetus volitans*; gills; marine; Lima (holotype, MUSM-HEL 283, paratype, author’s collection, 027 a-e) [150].

Remarks

The voucher specimens deposited in the author’s collection are not available. Status: valid species.


***Loxura peruensis* Oliva & Luque, 1995**


*Strongylura scapularis*; gills; marine; Lima (12°30′ S, 76°50′ W) (holotype, USNM 1377341; paratype, USNM 1376669, paratype, MUSM-HEL 246) [147].

Remarks

Status: valid species.


**Family Chauhaneidae Euzet & Trilles, 1960**



***Oaxacotyle oaxacensis* (Caballero & Bravo, 1964) Lebedev, 1984**


*Peprilus medius*; gills; marine; Lima (12°30′ S, 76°50′ W) (MUSM-HEL 2797) [151].

*Peprilus snyderi*; gills; marine; Lima (12°30′ S, 76°50′ W) (MUSM-HEL 3101) [152].

*Selene peruviana*; gills; marine; Tumbes (03°40′ S; 80°39′ W) (HPIA 189) [153].

Remarks

Status: valid species.


***Pseudochauhanea mexicana* Lamothe, 1966**


*Sphyraena ensis*; gills; marine; Tumbes (80°40′ S, 03°40′ W) (HPIA 196) [154].

*Sphyraena ensis*; gills; marine; Tumbes (80°40′ S, 03°40′ W) (HPIA 198a-p) [155].

Remarks

Referred to as *Pseudochauhanea* sp. by Minaya et al. [154]. Status: valid species.


***Pseudomazocraes* sp.**


*Seriola peruana*; gills; marine; Lambayeque (CPMP 505) (present study).


**Family Diclidophoridae Cerfontaine, 1895**



***Choricotyle anisotremi* Oliva, 1987**


*Anisotremus scapularis*; gills; marine; Lima (12°30′ S, 76°50′ W) (MUSM-HEL 2689) [156].

Remarks

Status: valid species.


***Choricotyle caulolatili* (Meserve, 1938) Sproston, 1946**


*Caulolatilus* sp.; gills; marine; Ancash (ITM 491, UNMSM-LPFSZ-91) [11].

*Caulolatilus affinis*; gills; marine; Piura (UNMSM-LPFSZ 437) [27].

*Caulolatilus princeps*; gills; marine; Lima [157].

Remarks

The voucher that was initially deposited in the IMT has been redeposited in the UNMSM-LPFSZ, as the IMT collection has been deactivated. Status: valid species.


***Choricotyle isaciencis* Oliva, González, Ruz & Luque, 2009**


*Isacia conceptionis*; gills; marine; Lima [158].

Remarks

No specimens in collections. Status: valid species.


***Choricotyle scapularis* Oliva, González, Ruz & Luque, 2009**


*Anisotremus scapularis*; gills; marine; Lima (CPMP 072) [26].

Remarks

Status: valid species.


***Choricotyle sonorensis* Caballero & Bravo, 1962**


*Isacia conceptionis*; gills; marine (UNMSM-LPFSZ-91) [142].

Remarks

*Choricotyle sonorensis* Caballero & Bravo, 1962 was considered a species inquirenda by Mamaev [159]. However, Tantaleán et al. [142] considered this species valid. Status: valid species.


***Hargicotyle chimbotensis* (Tantaleán, 1974) Mamaev & Aleshkina, 1984**


*Paralonchurus peruanus*; gills; marine; Lima (holotype, MUSM-HEL 281; paratypes in the author’s collection, M025a-f; paratypes UNMSM-LPFSZ-81) [10].

Remarks

The species was initially described as a member of the genus *Choricotyle* Van Beneden & Hesse, 1863, but was transferred to the genus *Hargicotyle* by Mamaev & Aleshkina [160]. The paratypes that were initially deposited in the IMT have been redeposited in the UNMSM-LPFSZ, as the IMT collection has been deactivated. Status: valid species.


***Hargicotyle magna* Oliva & Luque, 1989**


*Cheilotrema fasciatum*; gills; marine; Lima (12°30′ S, 76°50′ W) (holotype, MUSM-HEL 1030; paratypes, MUSM-HEL 1031; paratype, USNM 1375882; paratype, CHURP 507) [161].

*Stellifer minor*; gills; marine; Lima (12°30′ S, 76°50′ W) [88].

*Cheilotrema fasciatum*; gills, mouth; marine; Lima (12°30′ S, 76°50′ W) [89].

Remarks

The paratype deposited in the CHURP is not available because this collection has been deactivated. Status: valid species.


**Hargicotyle menticirrhi Oliva & Luque, 1989**


*Menticirrhus ophicephalus*; gills, mouth; marine; Lima (12°30′ S, 76°50′ W) (holotype, MUSM-HEL 1032; paratypes, MUSM-HEL 1033; paratype, USNM 1375880; paratype, CHURP 508) [161].

*Menticirrhus ophicephalus*; gills; marine; Lima (12°30′ S, 76°50′ W) [123].

*Menticirrhus ophicephalus*; gills; marine; Lima (12°30′ S, 76°50′ W) [88].

*Menticirrhus ophicephalus*; gills, mouth; marine; Lima [89].

Remarks

The paratype deposited in the CHURP is not available because this collection has been deactivated. Status: valid species.


***Hargicotyle paralonchuri* Oliva & Luque, 1989**


*Paralonchurus peruanus*; gills; marine; Lima (12°30′ S, 76°50′ W) (holotype, MUSM-HEL 1034; paratype, MUSM-HEL 1035; paratype, USNM 1375881; paratype, CHURP 508) [161].

*Paralonchurus peruanus*; gills; marine; Lima [87].

*Paralonchurus peruanus*; gills; marine; Lima (12°30′ S, 76°50′ W) [88].

*Paralonchurus peruanus*; gills, mouth; marine; Lima [89].

*Cynoscion analis*; gills; marine; Lima (12°30′ S, 76°50′ W) (MUSM-HEL 1736) [118].

Remarks

The paratype deposited in the CHURP is not available because this collection has been deactivated. Status: valid species.


***Hargicotyle peruensis* (Tantaleán, 1974) Mamaev & Aleshkina, 1984**


*Paralonchurus peruanus*; gills; marine; Lima (holotype, MUSM-HEL 280; paratypes in the author’s collection, M024a-g; paratypes UNMSM-LPFSZ-80) [10].

Remarks

The species was initially described as a member of the genus *Choricotyle* Van Beneden & Hesse, 1863 by Tantaleán [10], but was transferred to the genus *Hargicotyle* by Mamaev & Aleshkina [160]. The paratypes that were initially deposited in the IMT have been redeposited in the UNMSM-LPFSZ, as the IMT collection has been deactivated. Status: valid species.


***Hargicotyle sciaenae* Oliva & Luque, 1989**


*Sciaena deliciosa*; gills; marine; Lima (12°30′ S, 76°50′ W) [88].

*Sciaena deliciosa*; gills, mouth; marine; Lima [89].

*Sciaena deliciosa*; gills, opercule; marine; Lima (12°30′ S, 76°50′ S) [162].

*Sciaena deliciosa*; gills; marine; Lima (CPMP 003) [29].

*Sciaena deliciosa*; gills; marine; Lima (12°5′ S, 78°11′ W) (MUSM-HEL 3470a-h, CHIOC 38667a, b) [124].

*Sciaena deliciosa*; gills; marine; La Libertad [163].

Remarks

Chero et al. [29] recorded this species as *Hargicotyle louisiniensis* (Hargis, 1955) Mamaev, 1972, but a taxonomic study of the voucher specimens deposited in the CPMP showed that this record is conspecific with *H*. *scianae*. Chero et al. [124] redescribed this species. Status: valid species.


***Hargicotyle* sp.**


*Cilus gilberti*; gills; marine; Lima (12°30′ S, 76°50′ W) (CPMP 046) [164].

Remarks

This record was referred to as *Hargicotyle louisianensis* (Hargis, 1955) Mamaev, 1972 by Chero et al. [164], but a taxonomic study of the voucher specimens deposited in the CPMP showed that this record is not conspecific with *H*. *louisianensis* nor with any other species of *Hargicotyle* Mamaev, 1972.


***Hemitagia galapagensis* (Meserve, 1938) Sproston, 1946**


*Paralabrax humeralis*; gills; marine; Lima, La Libertad, Piura [16].

*Paralabrax humeralis*; gills; marine; Lima (12 30′ S, 76 50′ W) (MUSM-HEL 2710; 2716) [165].

Remarks

Status: valid species.


***Neoheterobothrium cynoscioni* (MacCallum, 1917) Price, 1943**


*Cynoscion analis*; gills; marine; Lima, Ica [16].

*Cynoscion analis*; gills; marine; Lima (12°30′ S, 76°50′ W) [118].

*Cynoscion analis*; gills; marine; Lima (12°30′ S, 76°50′ W) (CPYM 176) [119].

Remarks

Status: valid species.


***Olivacotyle hemanthiasi* Cruces, Chero, Sáez, Iannacone & Luque, 2017**


*Hemanthias signifer*; gills; marine; Tumbes (45°54′ S, 81°05′ W) (holotype, MUSM-HEL 3300; paratypes, MUSM-HEL 3301–3304; paratypes CHIOC 38882a-b) [166].

Remarks

*Hemanthicotyle sammarquensis* Luna, Martínez & Tantaleán, 2015 is a synonym of *Olivacotyle hemanthiasi* Cruces, Chero, Sáez, Iannacone & Luque, 2017. Status: valid species.


***Paraeurysorchis sarmientae* (Tantaleán, 1974) Tantaleán, Martinez & Escalante, 1985**


*Seriolella violacea*; gills; marine; Lima (holotype, MUSM-HEL 282, paratype in the author’s collection, M026 a-g) [12].

*Seriolella violacea*; gills; marine; Lima [16].

*Seriolella violacea*; gills; marine; Lima (11°52′ S, 77°07′ W) (MUSM-HEL 1751) [167].

*Seriolella violacea*; gills; marine; off Lima (USNM 1392386) (unpublished data, USNM).

Remarks

Tantaleán [12] proposed the genus *Pseudoeurysorchis* Tantaleán, 1974 to accommodate his new species, *Ps*. *sarmientoi*. However, the generic name had already been assigned to a diclidophorid species that infected haemulid fish from the Pacific Ocean, *Pseudoeurysorchis* Caballero & Bravo-Hollis, 1962. Thus, Tantaleán et al. [11] proposed the genus *Paraeurysorchis* Tantaleán, Martinez & Escalante, 1985 and transferred *Ps*. *sarmientoi* to *Paraeurysorchis*. The specific epithet represents a malformed suffix, and the correct name is *Paraeurysorchis sarmientae* [17]. Status: valid species.


***Pedocotyle annakohnae* Luque-Alejos & Iannacone-Oliver, 1989**


*Stellifer minor*; gills; marine; Lima (holotype, CHURP 501; paratype, CHIOC 32.494; paratype, CHURP 502–503) [168].

*Stellifer minor*; gills; marine; Lima (12°30′ S, 76°50′ W) [129].

*Stellifer minor*; gills; marine; Lima (12°30′ S, 76°50′ W) [88].

*Stellifer minor*; gills; marine; La Libertad [31].

*Stellifer minor*; gills; marine; Lima (12°S) [89].

Remarks

This species was described as *Pedocotyle annakohni* Luque & Iannacone, 1990, but the specific epithet represents a malformed suffix, and the correct name is *Pedocotyle annakohnae*. Currently, the CHURP is a deactivated collection, and all deposited material has been lost [17]. Status: valid species.


***Pedocotyle bravoi* Luque & Iannacone, 1989**


*Stellifer minor*; gills; marine; Lima (holotype, CHURP 504; paratypes, CNHE 242-12; paratype, CHURP 505-506) [168].

*Stellifer minor*; gills; marine; Lima (12°30′ S, 76°50′ W) [129].

*Stellifer minor*; gills; marine; Lima (12°30′ S, 76°50′ W) [88].

*Stellifer minor*; gills; marine; La Libertad [31].

*Stellifer minor*; gills; marine; Lima (12°S) [89].

Remarks

The voucher deposited in the CHURP is not available because this collection has been deactivated. Status: valid species.


***Pedocotyle* sp.**


*Stellifer minor*; gills; marine; Ica, Lambayeque [16].


***Pseudoeurysorchis travassosi* Caballero & Bravo-Hollis, 1962**


*Isacia conceptionis*; gills; marine; Lima (ITM 439) [11].

*Isacia conceptionis*; gills; marine; Ica [16].

*Isacia conceptionis*; gills; marine; Lima (CPMP 506) [159].

*Isacia conceptionis*; gills; marine; Lima (UNMSM-LPFSZ-92) (present study).

Remarks

The voucher specimens deposited in the IMT are not available. Status: valid species.


***Teleurysorchis gumercindae* Gonzales & Sarmiento, 1990**


*Schedophilus haedrichi*; gills; marine; Lima (09°05′ S, 78°33′ W) (holotype, MUSM-HEL 0540; paratypes, MUSM-HEL 0540a-b) [169].

Remarks

Status: valid species.


**Family Discocotylidae Price, 1936**



***Anthocotyle americana* (MacCallum, 1916) Price, 1943**


*Merluccius peruanus*; gills; marine; Lima [170].

*Merluccius peruanus*; gills; marine; Lima (11°45′ S, 77°58′ W) [171].

*Merluccius peruanus*; gills; marine; Lima [16].

*Merluccius peruanus*; gills; marine; Lima (CPMP 023–024) [172].

Remarks

Reported as *Anthocotyle americanus* by Duran and Oliva [170] and Verano et al. [171]. Status: valid species.


***Anthocotyle merluccii* Van Beneden & Hesse, 1863**


*Merluccius peruanus*; gills; marine; Lima [173].

Remarks

No specimens in collections. Status: valid species.


***Bicotylophora trachinoti* (MacCallum, 1921) Price, 1936**


*Trachinotus paitensis*; gills; marine; Ica, Lima, Piura [15,16].

Remarks

This record was considered in the checklists performed by Luque et al. [15] and Tantaleán & Huiza [16]. However, there are neither papers nor voucher specimens deposited in any collection that support the presence of this species in Peruvian waters. Status: valid species.


**Family Gastrocotylidae Price, 1943**



***Areotestis sibi* Yamaguti, 1965**


*Thunnus obesus*; nasal cavity; marine; Piura (05°04′ S, 81°06′ W) (UNMSM-LPFSZ 447) [32].

Remarks

Status: valid species.


***Pseudaxine* sp.**


*Scomberomorus sierra*; gills; marine; La Libertad [139].

Remarks

This record probably corresponds to *Mexicotyle mexicana* (Meserve, 1938) Lebedev, 1984, a common parasite of fish of the genus *Scomberomorus* Lacepède, 1801. However, there is no information on the scientific collection where the voucher specimens were deposited to corroborate this hypothesis. No specimens in collections.


**Family Heteraxinidae Unnithan, 1957**



***Allencotyla mcintoshi* Price, 1962**


*Seriola lalandi*; gills; marine; Ancash (IMT 562, UNMSM-LPFSZ-62) [142].

Remarks

The voucher that was initially deposited in the IMT has been redeposited in the UNMSM-LPFSZ, as the IMT collection has been deactivated.

Remarks

Status: valid species.


***Cemocotyle* aff. *trachuri* Dillon & Hargis, 1965**


*Trachurus murphyi*; gills; marine; La Libertad, Piura [16].

*Trachurus murphyi*; gills; marine; Piura [174].

Remarks

Referred to as *Cemocotyle* sp. by Tantaleán and Huiza [15]. No specimens in collections.


***Zeuxapta seriolae* (Meserve, 1938) Price, 1962**


*Chloroscombrus orqueta*; gills; marine; La Libertad, Lima (ITM 432, UNMSM-LPFSZ-32) [11].

*Chloroscombrus orqueta*; gills; marine; La Libertad, Lima [16].

Remarks

The voucher that was initially deposited in the IMT has been redeposited in the UNMSM-LPFSZ, as the IMT collection has been deactivated. Status: valid species.


**Family Hexostomatidae Price, 1936**



***Hexostoma lintoni* Price, 1936**


*Thunnus obesus*; gills; marine; Piura (05°04′ S, 81°06′ W) (UNMSM-LPFSZ 449) [32].

Remarks

Status: valid species.


**Family Macrovalvitrematidae Yamaguti, 1963**



***Macrovalvitrema* sp.**


*Sciaena deliciosa*; gills; marine; Lima (CPMP 004) [29].

*Cilus gilberti*; gills; marine; Lima (12°30’, 76°50′ W) (CPMP 045) [164].

Remarks

This record was erroneously identified as *Choricotyle* sp. for Chero et al. [164].


***Pterinotrematoides mexicanus* Caballero & Bravo-Hollis, 1955**


*Micropogonias* sp.; gills; marine; Lima (ITM 490, UNMSM-LPFSZ-90) [11].

Remarks

The voucher that was initially deposited in the IMT has been redeposited in the UNMSM-LPFSZ, as the IMT collection has been deactivated. Status: valid species.


**Family Mazocraeidae Price, 1936**



***Kuhnia indica* Tripathi, 1959**


*Scomber japonicus*; gills: marine; Arequipa [175].

Remarks

No specimens in collections. Status: valid species.


***Kuhnia* sp.**


*Trachinotus paitensis*; gills; marine; La Libertad [31].

Remarks

No specimens in collections. Erroneously referred to as Khunia sp. by Jara [31].


***Kuhnia scombri* (Kuhn, 1829) Sproston, 1945**


*Scomber japonicus*; gills; marine; Lambayeque (IMT 425) [142].

*Scomber japonicus*; gills; marine; Lima [16].

*Scomber japonicus*; gills: marine; Arequipa [175].

*Scomber japonicus*; gills; marine; Lima (12° S, 77° W) [176].

*Scomber japonicus*; gills; marine; Lima (UNMSM-LPFSZ-94) (present study).

Remarks

The voucher specimens deposited in the IMT are not available. No specimens of *K*. *scombri* from Arequipa were deposited in collections. Status: valid species.


***Kuhnia sprostonae* Price, 1961**


*Scomber japonicus*; gills; marine; Ancash, Lima (IMT 463) [142].

*Scomber japonicus*; gills; marine; Ancash, Lima [16].

*Scomber japonicus*; gills; marine; Lima (12° S, 77° W) [176].

*Scomber japonicus*; gills; marine; Lima (UNMSM-LPFSZ-96) (present study).

Remarks

The voucher specimens deposited in the IMT are not available. Referred to as *Kuhnis sprostonae* by Tantaleán et al. [142]. Status: valid species.


**Family Microcotylidae Taschenberg, 1879**



***Cynoscionicola americanus* Tantaleán, Martínez & Escalante, 1987**


*Menticirrhus ophicephalus*; gills; marine; Lima (UNMSM-LPFSZ 072) [177].

*Paralonchurus peruanus*; gills; marine; Lima [88].

*Menticirrhus ophicephalus*; gills; marine; Lima (12°30′ S, 76°50′ W) [123].

*Menticirrhus ophicephalus*, Paralonchurus peruanus; gills; marine; Lima (12°30′ S, 76°50′ W) [88].

Cheilotrema fasciatum, Paralonchurus peruanus, Stellifer minor, Menticirrhus ophicephalus; gills; marine; Lima [89].

*Cilus gilberti*; gills; marine; Lima (12°30′ S, 76°50′ W) (CPMP 047) (Chero et al. 2014d).

*Menticirrhus ophicephalus*; gills; marine; Lima (12°5′ S, 78°11′ W) (MUSM-HEL 3469a-e, CHIOC 38669a-b) [124].

Remarks

This species was described as *Cynoscionicola americana* Tantaleán, Martínez & Escalante, 1987 by Tantaleán et al. [177]. The specific epithet of this species was malformed. According to Chero et al. [178] and WoRMS [17], the correct name of this species is *C*. *americanus*. A redescription of this species was performed by Chero et al. [178]. Status: valid species.


***Cynoscionicola cynoscioni* Tantaleán, Martinez & Escalante, 1987**


*Cynoscion analis*; gills; marine; Lima (UNMSM-LPFSZ 070) [177].

*Stellifer minor*; gills; marine; Lima (12°30′ S, 76°50′ W) [88].

*Cynoscion analis*; gills; marine; Lima (12°30′ S, 76°50′ W) (MUSM-HEL 1737) [118].

*Cynoscion analis*; gills; marine; Lima (12°30’, 76°50’) [179].

*Cynoscion analis*; gills; marine; Lima (CPMP 174) [119].

Remarks

Status: valid species.


***Cynoscionicola intermedius* Tantaleán, Martinez & Escalante, 1988**


*Brachygenys peruanus*; gills; marine; Lima (holotype, MUSM-HEL 756, paratypes, IMT 654a-g) [142].

Remarks

This species was described as *Cynoscionicola intermedia* Tantaleán, Martínez & Escalante, 1988 by Tantaleán et al. [142]. The specific epithet of this species was malformed. According to WoRMS [17], the correct name of this species is *C*. *intermedius*. The paratypes deposited in the IMT are not available. Status: valid species.


***Cynoscionicola sciaenae* Tantaleán, 1974**


*Sciaena deliciosa*; gills; marine; Lima (holotype, MUSM-HEL 278, paratypes in the author’s collection, M022a-b) [13].

*Sciaena deliciosa*; gills; marine; Ica [16].

*Sciaena deliciosa*; gills; marine; Lima [89].

*Sciaena deliciosa*; gills; marine; Lima (12°30′ S, 76°50′ W) [162].

*Sciaena deliciosa*; gills; marine; Lima (CPMP 001) [29].

Remarks

The paratypes deposited in the author’s collection are not available. Status: valid species.


***Cynoscionicola* sp.**


*Stellifer minor*; gills; marine; Lima (12°30′ S, 76°50′ W) [129].

*Stellifer minor*; gills; marine; Lima (12°30′ S, 76°50′ W) [88].

Remarks

No specimens in collections.


***Cynoscionicola veranoi* Chero, Cruces, Saez & Luque, 2017**


*Sciaena deliciosa*; gills; marine; Lima (12°5′ S, 78°11′ W) (holotype, MUSM-HEL 3467, paratypes, MUSM-HEL 3468a-i, paratypes, CHIOC 38668a-d) [124].

Remarks

Status: valid species.


***Intracotyle neghmei* (Villalba, 1987) Oliva & Luque, 1995**


*Anisotremus scapularis*; gills; marine; Lima [147].

Remarks

*Intracotyle neghmei* (Villalba, 1987) Oliva & Luque, 1995 was described as *Neobivagina neghmei* Villalba, 1987 by Villalba [180], infecting the gills of *Anisotremus scapularis*. However, Olive and Luque [147] transferred this species to the genus *Intracotyle* Mamaev, 1970. Status: valid species.


***Jaliscia caballeroi* (Bravo-Hollis, 1960) Mamaev & Egorova, 1977**


*Caulolatilus* sp.; gills; marine; Lima [16].

*Caulolatilus affinis*; gills; marine; Piura (UNMSM-LPFSZ 438) [27].

*Caulolatilus princeps*; gills; marine; Lima (HPIA 32) [157].

*Mugil cephalus*; gills; marine; La Libertad [163].

Remarks

Referred to as *Microcotyle caballeroi* Bravo-Hollis, 1960 by Tantaleán and Huiza [16] and Luján and Ascón [163]. Status: valid species.


***Jaliscia* sp.**


*Caulolatilus affinis*; gills; marine; Piura (UNMSM-LPFSZ 439) [27].

*Caulolatilus princeps*; gills; marine; Lima (HPIA 31) [157].

Remarks

These records were mentioned as *Jaliscia caulolatili* Pérez-Urbiola, 1993. However, *J*. *caulolatili* is considered a nomen nudum since it was described in a bachelor thesis and never formally described according to the rules of the International Code for Zoological Nomenclature (ICZN). Thus, this monogenean needs to be formally described.


***Magniexcipula lamothei* Bravo-Hollis, 1981**


*Calamus brachysomus*; gills; marine; Tumbes (3°29′ S, 80°24′ W) (MUSM-HEL 4728a-f) [22].

Remarks

Status: valid species.


***Metamicrocotyla macracantha* (Alexander, 1954) Koratha, 1955**


*Mugil cephalus*; gills; marine; Lima [13].

*Mugil cephalus*; gills; marine; Ancash, Lima [16].

*Mugil cephalus*; gills; marine; Lima (12°30’ S, 76°50′ W) [181].

*Mugil cephalus*; gills; marine; Lima (12°30′ S, 76°50′ W) (MUSM-HEL 2019) [182].

*Mugil cephalus*; gills; marine; northern Peru [183].

*Mugil cephalus*; gills; marine; Lima (12°4′ S, 77°10′ W) (MUSM-HEL 3580) [184].

*Sphyraena ensis*; gills; marine; Tumbes (03°40′ S, 80°40′ W) (HPIA 197) [154].

*Selene peruviana*; gills; marine; Tumbes (03°40′ S, 80°39′ W) (HPIA 190) [153].

*Mugil cephalus*; gills; marine; La Libertad [163].

Remarks

Status: valid species.


***Microcotyle danielcarrioni* (Martinez & Barrantes, 1977) Bouguerche, Gey, Justine & Tazerouti, 2019**


*Cheilodactylus variegatus*; gills; marine; Lima (holotype, MUSM-HEL 126; paratypes, MUSM-HEL 127) [185].

Remarks

This species was designated as a member of the genus *Paramicrocotyle* by Martinez and Barrantes [185]. However, Bouguerche et al. [186] transferred *P*. *danielcarrioni* Martinez & Barrantes, 1977 to the genus *Microcotyle*. Oliva and Muñoz [187] considered *P*. *danielcarrioni* a synonym of *M*. *nemadactylus* Dillon & Hargis, 1965. However, *M*. *danielcarrioni* and *M*. *nemadactylus* are different species, according to Bouguerche et al. [186]. Status: valid species.


***Microcotyle nemadactylus* Dillon & Hargis, 1965**


*Cheilodactylus variegatus*; gills; marine; Lima [185].

*Cheilodactylus variegatus*; gills; marine; Lima (12°30’ S, 76°50′ W) [188].

*Cheilodactylus variegatus*; gills; marine; Lima (12°30’ S, 76°50′ W) [189].

*Cheilodactylus variegatus*; gills; marine; Lima (12°4′ S, 77°10′ W) (MUSM-HEL 3570) [184].

Remarks

This species was referred to as *Paramicrocotyle nemadactylus* (Dillon & Hargis, 1965) Caballero & Bravo-Hollis, 1972 by Iannacone et al. [189]. Status: valid species.


***Microcotyle* sp.**


*Hyporthodus niphobles*; gills; marine; Lima (12°28′ S, 76°47′ W) (HPIA 143) [20].


***Microcotyloides impudicus* Caballero, Bravo-Hollis & Grocott, 1954**


*Polydactylus approximans*; gills; marine; Lima [16].

Remarks

Status: valid species.


***Neobivagina chita* Tantaleán, Morales & Escalante, 1998**


*Anisotremus scapularis*; gills; marine; Ancash, Ica, Lima (holotype, MUSM-HEL 1056, paratypes, MUSM-HEL 1057–1058) [148].

*Anisotremus scapularis*; gills; marine; Lima (12°30’ S, 76°50′ W) (MUSM-HEL 2707) [156].

*Anisotremus scapularis*; gills; marine; Lima (CPMP 073) [26].

Remarks

The type locality of this species was not mentioned in its original description. Status: valid species.


**Family Protomicrocotylidae Johnston & Tiegs, 1922**



***Neomicrocotyle* sp.**


*Caranx hippos*; gills; marine; La Libertad, Piura [16].


**Family Pyragraphoridae Yamaguti, 1963**



***Pyragraphorus pyragraphorus* (MacCallum & MacCallum, 1913) Sproston, 1946**


*Trachinotus paitensis*; gills; marine; Piura (05°05′ S, 81°07′ W) (CHURP 596) [14].

Remarks

The voucher deposited in the CHURP is not available because this collection has been deactivated. Status: valid species.


**Family Thoracocotylidae Price, 1936**



***Mexicotyle mexicana* (Meserve, 1938) Lebedev, 1984**


*Scomberomorus sierra*; gills; marine; Ica [139].

*Scomberomorus sierra*; gills; marine; Ancash (IMT 561, UNMSM-LPFSZ-61) [142].

*Scomberomorus sierra*; gills; marine; Ica [16].

*Scomberomorus sierra*; gills; marine; Tumbes (13°30′ S, 80°24′ W) (HPIA 211) [190].

*Scomberomorus sierra*; gills; marine; Lima (12°5′ S, 78°11′ W) (CPMP 507) (new geographical extension) (present study).

*Scomberomorus sierra*; gills; marine; Tumbes (3°29′ S, 80°24′ W) (CPMP 508) (present study).

Remarks

This species was registered in Peru as *Pseudaxine* sp. by Escalante et al. [139] and later identified as *Pseudaxine mexicana* Meserve, 1938 by Tantaleán et al. [142]. However, *P*. *mexicana* was transferred to the genus *Mexicotyle* Lebedev, 1984 by Lebedev [191]. The voucher that was initially deposited in the IMT has been redeposited in the UNMSM-LPFSZ, as the IMT collection has been deactivated. Status: valid species.


***Scomberocotyle scomberomori* (Koratha, 1955) Harguis, 1956**


*Sphyraena ensis*; gills; marine; Tumbes (80°40′ S, 03°40′ W) (HPIA 198) [154].

*Sphyraena ensis*; gills; marine; Tumbes (80°40′ S, 03°40′ W) (HPIA 198a-b) [155].

*Scomberomorus sierra*; gills; marine; Tumbes (13°30′ S, 80°24′ W) (HPIA 212) [190].

*Scomberomorus sierra*; gills; marine; Lima (12°5′ S, 78°11′ W) (CPMP 509) (new geographical extension) (present study).

*Scomberomorus sierra*; gills; marine; Tumbes (3°29′ S, 80°24′ W) (CPMP 510) (present study).

Remarks

Referred to as *Scomberocotyle* sp. by Minaya et al. [154]. Status: valid species.


***Thoracocotyle crocea* MacCallum, 1913**


*Scomberomorus sierra*; gills; marine; La Libertad (IMT 433, UNMSM-LPFSZ-33) [11].

*Scomberomorus sierra*; gills; marine; Ica [16].

*Scomberomorus sierra*; gills; marine; Tumbes (13°30′ S, 80°24′ W) (HPIA 210) [190].

*Scomberomorus sierra*; gills; marine; Lima (12°5′ S, 78°11′ W) (CPMP 511) (new geographical extension) (present study).

*Scomberomorus sierra*; gills; marine; Tumbes (3°29′ S, 80°24′ W) (CPMP 512) (present study).

Remarks

The voucher that was initially deposited in the IMT has been redeposited in the UNMSM-LPFSZ, as the IMT collection has been deactivated. Status: valid species.


**Order Polystomatidea Lebedev, 1988**



**Family Polystomatidae Gamble, 1896**



***Mesopolystoma samiriensis* Vaucher, 1981**


*Osteocephalus taurinus*; urinary bladder; terrest; Loreto (holotype, MHNG 980.472) [192].

Remarks

The description of this species was based on a single specimen. Status: valid species.


***Polistoma* sp.**


*Dendropsophus pauiniensis*; urinary bladder; terrest; Madre de Dios (unpublished data, USNM 1377686, 1377687).


***Wetapolystoma almae* Gray, 1993**


*Rhinella margaritifera* (type host); urinary bladder; terrest; Madre de Dios (holotype, USNM 1377688) [193].

Remarks

The description of this species was based on a single specimen. Status: valid species.

### 3.2. Host–Parasite List


**Phylum Chordata Haeckel, 1874**



**Class Teleostei Müller, 1845**



**Order Acanthuriformes Jordan, 1923**



**Family Ephippidae Bleeker, 1859**



***Parapsettus panamensis* (Steindachner, 1876**


Sprostoniella lamothei, Parancylodiscoides chaetodipteri


**Order Anabantiformes Britz, 1995**



**Family Osphronemidae van der Hoeven, 1832**



***Trichopodus trichopterus* Pallas, 1770**


Trianchoratus acleithrium


**Order Beloniformes Berg, 1937**



**Family Belonidae Bonaparte, 1835**



***Strongylura scapularis* Jordan & Gilbert, 1882**


Tylosuricola amatoi, Loxura peruensis


**Family Exocoetidae Risso, 1827**



***Exocoetus volitans*
Linnaeus, 1758**


Axine ibanezi


**Order Eupercaria Betancur-R et al., 2013 incertae sedis**



**Family Haemulidae Gill, 1885**



***Anisotremus scapularis*
Tschudi, 1846**


Choricotyle anisotremi, C. scapularis, Encotyllabe antofagastensis, Intracotyle neghmei, Mexicana sp., Neobivagina chita


***Brachygenys peruanus*
Hildebrand, 1946**


Cynoscionicola intermedius


***Haemulon steindachneri* Jordan & Gilbert, 1882**



*Mexicana iannaconi*



***Isacia conceptionis* Cuvier, 1830**


*Benedenia* sp., *Choricotyle isaciencis*, *C. sonorensis*, *Pseudoeurysorchis travassosi*


***Orthopristis chalceus* Günther, 1864**


*Calceostoma* sp.


**Family Labridae Cuvier, 1816**



***Bodianus diplotaenia* Gill, 1862**


*Haliotrema diplotaenia*, *H. saezae*


**Family Malacanthidae Poey, 1861**



***Caulolatilus* sp.**



*Choricotyle caulolatili*



***Caulolatilus affinis* Gill, 1865**


*Encotyllabe pagrosomi*, *Choricotyle caulolatili*, *Jaliscia* sp., *J. caballeroi*


***Caulolatilus princeps* Jenyns, 1840**


*Choricotyle caulolatili*, *Jaliscia* sp., *J. caballeroi*


**Family Scaridae Rafinesque, 1810**



***Scarus perrico* Jordan & Gilbert, 1882**



*Haliotrema sanchezae*



**Family Sciaenidae Cuvier, 1829**



***Cheilotrema fasciatum* Tschudi, 1846**


*Cynoscionicola americanus*, *Hargicotyle magna*, Diplectanidae gen. sp1., Diplectanidae gen. sp2., Diplectanum sp. 2., Diplectanum sp. 3., *Rhamnocercus fasciatus*


***Cilus gilberti* Abbott, 1899**


*Cynoscionicola americanus*, *Hargicotyle* sp., *Macrovalvitrema* sp.


***Cynoscion analis* Jenyns, 1842**


*Cynoscionicola cynoscioni*, *Diplectanum* sp. 1., *Hargicotyle paralonchuri*, *Neoheterobothrium cynoscioni*


***Cynoscion phoxocephalus* Jordan & Gilbert, 1882**



*Cynoscionella sanmarci*



***Menticirrhus elongatus* Günther, 1864**



*Rhamnocercoides lambayequensis*



***Menticirrhus ophicephalus* Jenyns, 1840**


*Cynoscionicola americanus*, *Hargicotyle menticirrhi*, *Rhamnocercoides menticirrhi*


***Micropogonias* sp.**



*Pterinotrematoides mexicanus*



***Paralonchurus peruanus* Steindachner, 1875**


*Euryhaliotrema paralonchuri*, Diplectanidae gen. sp3., Diplectanum sp. 4., *Rhamnocercus dominguesi*, *Hargicotyle chimbotensis*, *H. paralonchuri*, *H. peruensis*, *Cynoscionicola americanus*


***Pareques lanfeari* Barton, 1947**


*Rhamnocercus chacllae*, *R. chaskae*


***Sciaena deliciosa* Tschudi, 1846**


*Cynoscionicola sciaenae*, *Cynoscionicola veranoi*, *Encotyllabe* sp1., *Hargicotyle sciaenae*, *Macrovalvitrema* sp.


***Umbrina xanti* Gill, 1862**



*Euryhaliotrema sagmatum*



**Family Sparidae Rafinesque, 1818**



***Calamus brachysomus* Lockington, 1880**


*Euryhaliotrema luisae*, *E. magnopharyngis*, *Haliotrematoides mediohamides*, *H. prolixohamus*, *Magniexcipula lamothei*


**Order Carangaria Betancur-R et al., 2013 incertae sedis**



**Family Sphyraenidae Rafinesque, 1815**



***Sphyraena ensis* Jordan & Gilbert, 1882**


*Metamicrocotyla macracantha*, *Pseudochauhanea mexicana*, *Scomberocotyle scomberomori*


**Family Polynemidae Rafinesque, 1815**



***Polydactylus approximans* Lay & Bennett, 1839**



*Microcotyloides impudicus*



**Order Carangiformes Jordan, 1923**



**Family Carangidae Rafinesque, 1815**



***Caranx hippos* Linnaeus, 1766**


*Allopyragraphorus caballeroi*, *Neomicrocotyle* sp.


***Chloroscombrus orqueta* Jordan & Gilbert, 1883**



*Zeuxapta seriolae*



***Selene peruviana* Guichenot, 1866**


*Metamicrocotyla macracantha*, *Oaxacotyle oaxacensis*


***Seriola peruana* Steindachner, 1881**



***Pseudomazocraes* sp.**



***Trachurus murphyi* Nichols, 1920**



*Cemocotyle aff. trachuri*



***Trachinotus paitensis* Cuvier, 1832**


*Bicotylophora trachinoti*, *Pyragraphorus pyragraphorus*, *Kuhnia* sp.


***Seriola lalandi* Valenciennes, 1833**



*Allencotyla mcintoshi*



**Order Centrarchiformes Bleeker, 1859**



**Family Cheilodactylidae Bonaparte, 1850**



***Cheilodactylus variegatus* Valenciennes, 1833**


Capsalidae gen. sp., *Encotyllabe cheilodactyli*, *Microcotyle danielcarrioni*, *Microcotyle nemadactylus*


***Plagioscion squamosissimus* Heckel, 1840**


*Euryhaliotrema chaoi*, *E. lovejoyi*, *E. monacanthus*, *E. potamocetes*, *E. succedaneus*, *E. thatcheri*, *Diplectanum decorum*, *D. pescadae*, *D. piscinarius*


***Stellifer minor* Tschudi, 1846**


*Cynoscionicola* sp., *Cynoscionicola americanus*, *C. cynoscioni*, *Encotyllabe* sp2., *Hargicotyle magna*, *Loimos* sp., *Pedocotyle annakohnae*, *P. bravoi*, *Rhamnocercus* sp., *R. oliveri*, *R. rimaci*, *R. stelliferi*, *R. tantaleani*.


**Order Characiformes Goodrich, 1909**



**Family Bryconidae Eigenmann, 1912**



***Brycon amazonicus* Spix & Agassiz, 1829**


*Anacanthorus femoris*, *A. kukamensis*, *A. rarus*, *A. sabaloi*, *A. spiralocirrus*, *Jainus peruensis*, *Notozothecium agusti*


***Brycon cephalus* Günther, 1869**



*Jainus amazonensis*



**Family Characidae Latreille, 1825**



***Eretmobrycon peruanus* Müller & Troschel, 1845**



***Anacanthocotyle* sp.**



**Family Lebiasinidae Gill, 1889**



***Lebiasina bimaculata* Valenciennes, 1847**


*Accessorius peruensis*, *Gyrodactylus bimaculatus*, *G. lebiasinus*, *G. slendrus*, *Onchocleidus* sp.


**Family Prochilodontidae Eigenmann, 1909**



***Prochilodus nigricans* Spix & Agassiz, 1829**


*Apedunculata discoidea*, *Rhinonastes pseudocapsaloideum*, *Tereancistrum curimba*, *T. toksonum*


**Family Triportheidae Fowler, 1940**



***Triportheus angulatus* Spix & Agassiz, 1829**


*Anacanthorus acuminatus*, *A. chaunophallus*, *A chelophorus*, *A. euryphallus*, *A. lygophallus*, *A. pithophallus*, *Ancistrohaptor falciferum*, *A. falcunculum*, *Jainus* sp1., *Jainus* sp2., *Jainus* sp3.


**Family Serrasalmidae Bleeker, 1859**



***Colossoma macropomum* Cuvier, 1816**


*Anacanthorus* sp2., *A. spathulatus*, *Dactylogyridae gen.* sp1., *Dactylogyridae gen.* sp9., *Dactylogyrus* sp., *Gyrodactylidae gen.* sp., *Mymarothecium* sp., *M. boegeri*, *M. iiapensis*, *M. viatorum*, *M. tantaliani*, *Notozothecium janauachensis*


***Myloplus schomburgkii* Jardine, 1841**


*Anacanthorus* sp1., *A. camposbaci*, *A. carmenrosae*, *A. pedanophallus*, *Notozothecium* sp2., *N. bethae*, *N. nanayense*


***Piaractus brachypomus* Cuvier, 1818**


*Anacanthorus penilabiatus*, *Mymarothecium viatorum*


***Pygocentrus nattereri* Kner, 1858**


*Amphithecium calycinum*, *A. camelum*, *A. cataloensis*, *A. falcatum*, *A. junki*, *Anacanthorus amazonicus*, *A. penilabiatus*, *A. ramosissimus*, *A. reginae*, *A. scapanus*, *A. stachophallus*, *A. stachophallus*, *A. thatcheri*, *Enallothecium aegidatum*, *Mymarothecium galeolum*, *M. viatorum*, *Notothecioides llewellyni*, *Notozothecium* sp1., *N. minor*, *N. mizellei*, *N. penetrarum*, *Rhinoxenus piranhus*, *Urocleidoides eremitus*


**Order Cichliformes Betancur-R et al., 2013**



**Family Cichlidae Bonaparte, 1835**



***Acaronia nassa* Heckel, 1840**


*Gussevia* sp1., *Gussevia* sp2., *Gussevia* sp3.


***Aequidens tetramerus* Heckel, 1840**


*Gussevia cichlasomatis*, *Gussevia* sp4., *Gussevia* sp5., *Gussevia* sp6., *Sciadicleithrum* sp1.


***Andinoacara rivulatus* Günther, 1860**


*Urocleidus* sp.


***Apistogramma* sp.**


*Dactylogyrus* sp.


***Astronotus ocellatus* Agassiz, 1831**


*Gussevia asota*, *G. astronoti*, *G. rogersi*


***Biotodoma cupido* Heckel, 1840**


*Biotodomella mirospinata*, *Sciadicleithrum* sp2.


***Bujurquina peregrinabunda* Kullander, 1986**


*Sciadicleithrum* sp3., *Sciadicleithrum* sp4.


***Chaetobranchus flavescens* Heckel, 1840**



*Sciadicleithrum edgari*



***Chaetobranchus semifasciatus* Steindachner, 1875**



*Gussevia tucunarensis*



***Cichla monoculus* Agassiz, 1831**


*Gussevia arilla*, *G. longihaptor*, *G. tucanarensis*, *G. undulata*, *Sciadicleithrum ergensi*, *S. umbilicum*, *S. uncinatum*, *Tucunarella cichlae*


***Cichlasoma amazonarum* Kullander, 1983**


*Gussevia alii*, *G. cichlasomatis*, *G. disparoides*, *Gussevia* sp7., *Gussevia* sp8., *Gussevia* sp10., *Gussevia* sp11., *Sciadicleithrum variabile*, *Trinidactylus cichlasomatis*


***Cichlasoma* sp.**


*Dactylogyrus* sp.


***Crenicichla johanna* Heckel, 1840**


*Sciadicleithrum* sp5., *Sciadicleithrum* sp6., *Sciadicleithrum* sp7.


***Heros efasciatus* Heckel, 1840**


*Gussevia* sp9., *G. dispar*, *G. disparoides*, *Sciadicleithrum variabile*


***Heros severus* Heckel, 1840**


*Gussevia alioides*,


***Mesonauta festivus* Heckel, 1840**


*Gussevia* sp12., *Sciadicleithrum* sp9.


***Mesonauta mirificus* Kullander & Silfvergrip, 1991**


*Sciadicleithrum* sp8.

*Oreochromis* sp.

*Gyrodactylus* sp3.

Oreochromis niloticus Linnaeus, 1758

*Cichlidogyrus* sp., *C. sclerosus*, *C. tilapiae*


***Pterophyllum scalare* Schultze, 1823**


*Dactylogyrus* sp., *Gussevia spiralocirra*, *Sciadicleithrum iphthimum*


***Satanoperca jurupari* Heckel, 1840**



*Sciadicleithrum satanopercae*



***Symphysodon aequifasciatus* Pellegrin, 1904**


*Dactylogyrus* sp.


***Symphysodon tarzoo* Lyons, 1959**



*Sciadicleithrum variabile*



**Order Cypriniformes Goodrich, 1909**



**Family Cyprinidae Rafinesque, 1815**



***Carassius auratus* Linnaeus, 1758**


*Gyrodactylus* sp., *Dactylogyrus* sp.


***Cyprinus carpio* Linnaeus, 1758**



*Dactylogyrus vastator*



**Order Cyprinodontiformes Parenti, 1981**



**Family Poeciliidae Bonaparte, 1831**



***Poecilia reticulata* Peters, 1859**



*Gyrodactylus turnbulli*



**Order Gadiformes Goodrich, 1909**



**Family Merlucciidae Rafinesque, 1815**



***Merluccius peruanus* Ginsburg, 1954**


*Anthocotyle americana*, *A. merluccii*


**Order Mugiliformes Günther, 1880**



**Family Mugilidae Jarocki, 1822**



***Mugil cephalus* Linnaeus, 1758**


*Ligophorus mugilinus*, *Neobenedia* sp., *Neobenedia pacifica*, *Pseudohaliotrema* sp., *Metamicrocotyla macracantha*


**Order Ophidiiformes Berg, 1937**



**Family Ophidiidae Rafinesque, 1810**



**Ophidiidae not identified**



***Neobenedia* sp.**



***Brotula clarkae* Hubbs, 1944**


*Brotulella laurafernandae*, *B. luisahelenae*


**Order Osteoglossiformes Greenwood et al., 1966**



**Family Arapaimidae Bonaparte, 1846**



***Arapaima gigas* Schinz, 1822**


*Dawestrema cycloancistrioides*, *D. cycloancistrium*


**Family Osteoglossidae Bonaparte, 1845**



***Osteoglossum bicirrhosum* Cuvier, 1829**


*Gonocleithrum aruanae*, *G. coenoideum*, *G. cursitans*


**Order Perciformes Johnson, 1984**



**Family Serranidae Swainson, 1839**



***Epinephelus labriformis* Jenyns, 1840**



*Pseudorhabdosynochus anulus*



***Hemanthias peruanus* Steindachner, 1875**


*Bicentenariella peruensis*, *Pronotogrammella scholzi*


***Hemanthias signifer* Garman, 1899**


*Bicentenariella signiferi*, *Olivacotyle hemanthiasi*


***Hyporthodus niphobles* Gilbert & Starks, 1897**


*Benedenia* sp., *Microcotyle* sp.


***Paralabrax humeralis* Valenciennes, 1828**



*Hemitagia galapagensis*



***Paranthias colonus* Valenciennes, 1846**



*Pseudorhabdosynochus jeanloui*



***Pronotogrammus multifasciatus* Gill, 1863**


*Bicentenariella claudiae*, *B. puertopizarroensis*, *B. sinuosa*, *Pronotogrammella boegeri*, *P. multifasciatus*, *P. scholzi*


**Order Scombriformes Johnson, 1986**



**Family Centrolophidae Bonaparte, 1846**



***Schedophilus haedrichi* Chirichigno F., 1973**



*Teleurysorchis gumercindae*



***Seriolella violacea* Guichenot, 1848**



*Paraeurysorchis sarmientae*



**Family Scombridae Rafinesque, 1815**



***Thunnus albacares* Bonnaterre, 1788**


*Capsala biparasitica*, *Nasicola klawei*


***Thunnus obesus* Lowe, 1839**


*Capsala paucispinosa*, *Nasicola klawei*, *Areotestis sibi*, *Hexostoma lintoni*


***Sarda chiliensis* Cuvier, 1832**



*Capsala gregalis*



***Scomber japonicus* Houttuyn, 1782**


*Kuhnia* sp., *K. indica*, *K. scombri*, *K. sprostonae*


***Scomberomorus sierra* Jordan & Starks, 1895**


*Mexicotyle mexicana*, *Pseudaxine* sp., *Scomberocotyle scomberomori*, *Thoracocotyle crocea*


**Family Stromateidae Rafinesque, 1810**



***Peprilus medius* Peters, 1869**



*Oaxacotyle oaxacensis*



***Peprilus snyderi* Gilbert & Starks, 1904**



*Oaxacotyle oaxacensis*



**Order Siluriformes Hay, 1929**



**Family Auchenipteridae Bleeker, 1862**



***Ageneiosus vittatus* Steindachner, 1908**


*Boegeriella conica*, Dactylogyridae gen. sp2.


***Centromochlus heckelii* De Filippi, 1853**



*Demidospermus centromochli*


***Trachelyopterus*** sp.

*Cosmetocleithrum baculum*, *C. laciniatum*


**Family Ariidae Bleeker, 1858**



***Galeichthys peruvianus* Lütken, 1874**


*Hamatopeduncularia* sp.


**Family Callichthyidae Bonaparte, 1835**



***Corydoras ambiacus* Cope, 1872**


*Philocorydoras alcantarai*, *P. beleniensis*


***Corydoras multiradiatus* Orcés V., 1960**


*Philocorydoras jumboi*, *P. multiradiatus*


***Corydoras splendens* Castelnau, 1855**


*Philocorydoras maltai*, *P. peruensis*


**Family Doradidae Bleeker, 1858**



***Anadoras grypus* Cope, 1872**



*Cosmetocleithrum infinitum*



***Hassar* sp.**


*Ameloblastella* sp1.


***Hassar orestis* Steindachner, 1875**



*Cosmetocleithrum bifurcum*



***Nemadoras hemipeltis* Eigenmann, 1925**



*Cosmetocleithrum tortum*



***Megalodoras uranoscopus* Eigenmann & Eigenmann, 1888**



*Cosmetocleithrum falsunilatum*



***Oxydoras niger* Valenciennes, 1821**


*Cosmetocleithrum confusum*, *C. gigas*, *C. gussevi*, *C. parvum*, *C. rarum*, *C. sobrinus*


***Pterodoras granulosus* Valenciennes, 1821**


*Cosmetocleithrum bulbocirrus*, *Vancleaveus janauacaensis*


**Family Heptapteridae Gill, 1861**



***Goeldiella eques* Müller & Troschel, 1849**


*Aphanoblastella* sp., *A. aurorae*


***Pimelodella yuncensis* Steindachner, 1902**


*Cleidodiscus* sp., *Gyrodactylus pimelodellus*, *Scleroductus yuncensi*


**Family Loricariidae Rafinesque, 1815**



***Farlowella amazonum* Günther, 1864**



*Oogyrodactylus farlowellae*



***Loricaria* sp.**



*Demidospermus wilberi*



***Pterygoplichthys anisitsi* Eigenmann & Kennedy, 1903**


*Heteropriapulus heterotylus*, *Unilatus brittani*, *U. unilatus*


***Pterygoplichthys pardalis* Castelnau, 1855**


*Unilatus unilatus*, *U. brittani*, *Trinigyrus peregrinus*


**Family Pimelodidae Bonaparte, 1835**



**Pimelodidae gen. sp.**



*Ameloblastella formatrium*



***Aguarunichthys torosus* Stewart, 1986**



*Unibarra paranoplatensis*



***Brachyplatystoma juruense* Boulenger, 1898)**


*Boegeriella conica*, *Demidospermus mortenthaleri*


***Brachyplatystoma tigrinum* Britski, 1981**



*Peruanella madredediosensis*



***Brachyplatystoma vaillantii* Valenciennes, 1840**


*Demidospermus* sp1., *Demidospermus* sp2.


***Calophysus macropterus* Lichtenstein, 1819**


*Ameloblastella unapi*, *Demidospermus* sp3., *D. macropteri*


***Hemisorubim platyrhynchos* Valenciennes, 1840**


*Ameloblastella martinae*, *Vancleaveus platyrhynchi*


***Hypophthalmus* sp.**



*Ameloblastella peruensis*



***Hypophthalmus edentatus* Spix & Agassiz, 1829**


*Ameloblastella* sp2., *A. edentensis*, Dactylogyridae gen. sp5., Dactylogyridae gen. sp6., Dactylogyridae gen. sp7.


***Pimelodus* sp.**


*Ameloblastella unapioides*, *Demidospermus brevicirrus*, *D. curvovaginatus*, *D. peruvianus*, *D. striatus*


***Pimelodus blochii* Valenciennes, 1840**


*Demidospermus peruvianus*, *D. striatus*


***Pimelodus ornatus* Kner, 1858**



*Demidospermus peruvianus*



***Platynematichthys notatus* Jardine, 1841**


*Boegeriella conica*, Dactylogyridae gen. sp3, Dactylogyridae gen. sp8.


***Platystomatichthys sturio* Kner, 1858**



*Boegeriella ophiocirrus*



***Pseudoplatystoma fasciatum* Linnaeus, 1766**



*Vancleaveus fungulus*



***Pseudoplatystoma punctifer* Castelnau, 1855**


*Ameloblastella martinae*, *Demidospermus doncellae*, *Mymarothecium* sp., *Nanayella* sp., *Peruanella aureagarciae*, *Vancleaveus cicinnus*, *V. fungulus*


***Sorubim lima* Bloch & Schneider, 1801**


*Ameloblastella martinae*, *A. unapioides*, Dactylogiridae gen. sp4., *Nanayella aculeatrium*, *N. megorchis*


**Family Trichomycteridae Bleeker, 1858**



***Trichomycterus punctulatus* Valenciennes, 1846**


*Gyrodactylus* sp.


**Class Elasmobranchii Müller, 1845**



**Order Carcharhiniformes Garman, 1913**



**Family Triakidae Gray, 1851**



***Mustelus* sp.**


*Erpocotyle* sp.


***Mustelus dorsalis* Gill, 1864**



*Loimos scoliodoni*



***Triakis maculata* Kner & Steindachner, 1867**


*Erpocotyle* sp.


**Order Chimaeriformes Patterson, 1965**



**Family Callorhinchidae Garman, 1901**



***Callorhinchus callorynchus* Linnaeus, 1758**


*Callorhynchicola branchialis*, *Callorhynchocotyle callorhynchi*, *C. marplatensis*


**Order Myliobatiformes Compagno, 1973**



**Family Dasyatidae Jordan & Gilbert, 1879**



***Hypanus dipterurus* Jordan & Gilbert, 1880**


*Heterocotyle margaritae*, *Hypanocotyle bullardi*, *Listrocephalos kearni*, *Loimopapillosum pascuali*, *Monocotyle luquei*, *Peruanocotyle chisholmae*


**Family Myliobatidae Bonaparte, 1835**



***Myliobatis peruvianus* Garman, 1913**


*Benedeniella* sp.


**Family Potamotrygonidae Garman, 1877**



**Potamotrygon motoro Müller & Henle, 1841**


*Potamotrygonocotyle chisholmae*, *P. dromedarius*


***Potamotrygon* sp.**


*Potamotrygonocotyle rionegrense*, *P. tsalickisi*


**Order Rhinopristiformes Naylor et al., 2012**



**Family Rhinobatidae Bonaparte, 1835**



***Pseudobatos planiceps* Garman, 1880**


*Anoplocotyloides chorrillensis*, *A. papillatus*, *Heterocotyle* sp., *Rhinobatonchocotyle pacifica*


**Class Amphibia Linnaeus, 1758**



**Orden Anura Duméril, 1806**



**Family Bufonidae Gray, 1825**



***Rhinella margaritifera* Laurenti, 1768**



*Wetapolystoma almae*



**Family Hylidae Rafinesque, 1815**



***Dendropsophus pauiniensis* Heyer, 1977**


*Polistoma* sp.


***Osteocephalus taurinus* Steindachner, 1862**



*Mesopolystoma samiriensis*


## 4. Discussion

The checklist presented herein includes 358 taxa of monogeneans, comprising 270 nominal species and 88 taxa identified across different hierarchical levels, found in 145 host species. Among these, 335 taxa have been reported from bony fishes, 20 from cartilaginous fish and three species, namely *Mesopolystoma samiriensis*, *Polistoma* sp. and *Wetapolystoma almae*, found in amphibian hosts. The diversity of recorded monogeneans in Peru is comparable to that found in other countries across South and Central America, such as Brazil and Mexico, with 471 and 367 taxa, respectively [9,194]. However, this diversity is lower than in other regions of the world, such as in China, which boasts 581 species of monogeneans from aquatic vertebrates. Given the diverse spectrum of aquatic hosts inhabiting Peru, including freshwater fish, marine fish, and amphibians, there exists a conducive environment to support a rich diversity of monogenean species. Monogeneans, known for their exceptional specialization among parasites, demonstrate a unique affinity for specific host species. This specialization often results in a high degree of host specificity, suggesting that the potential diversity of monogenean species in Peru could be substantial. Out of the 10 orders of monogeneans recorded worldwide, 8 of them have been documented in Peru. The order with the highest number of species is Dactylogyroidea Bychowsky, 1933, which includes, among others, monogeneans from the families Dactylogyridae Bychowsky, 1933 and Diplectanidae Monticelli, 1903 (Figure 1). These families are recognized for their extensive diversity in the Neotropical region. The Dactylogyridae, with 50 genera and 217 species, is the most diverse family in Peru (Figure 1). This family comprises around 166 genera and over 1000 species worldwide [195]. Consequently, in Peru, nearly one-fourth of the known genera are reported, indicating the high diversity of this family in the country. Freshwater dactylogyrids exhibited the highest number of species. Among these, members of the genus *Anacanthorus* Mizelle & Price, 1965 are notably the most extensively studied group, primarily parasitizing characiformes fish in the northeastern Peruvian Amazon. In relation to the Diplectanidae, 24 taxa parasitizing fish in Peru have been documented, with the genus *Rhamnocercus* Monaco, Wood & Mizelle, 1954 being the most diverse in terms of the number of species. In fact, out of the 12 currently recognized *Rhamnocercus* species, 8 are recorded in Peru [125]. The diversity of dactylogyroid monogeneans is owed largely to the endeavors of specific researchers who commenced a series of taxonomic studies in recent years focusing on both marine [57,125] and freshwater dactylogyroids [42,82,83,102].

The initial advancement in understanding monogeneans in marine fish from the Peruvian Sea (southeastern Pacific Ocean) was conducted by Dr. Manuel Tantaléan Vidaurre (†) in 1973 (Figure 2).

This study formed a part of his doctoral thesis titled “Nuevo género y nuevas especies de monogeneos parásitos de peces de algunas áreas de la costa peruana”. Within this research, seven species of monogeneans were identified, among which six were newly discovered species and one belonged to a previously unknown genus in the scientific community. Since then, investigations into this crucial group of parasites in Peru have persisted until the present day. Iannacone and Luque [14] made the pioneering effort to collate information on monogenean records in the country, concentrating solely on those associated with marine fish. Their list documented a total of 37 species, with only two, at that juncture, recorded as infecting elasmobranch fish. However, it is crucial to note that certain records considered in the work of Iannacone & Luque [14] lacked support from formal publications (merely presented at scientific meetings), and in some instances, like that of *Bicotylophora trachinoti*, there was an absence of material deposited in a scientific collection. Cohen et al. [196] increased the understanding of monogeneans in Peru by compiling a list that encompassed monogeneans infecting both freshwater and marine fish, along with amphibians. By that point, 115 monogeneans had been identified in Peru. The most recent inventory of monogeneans in marine and freshwater fish from Peru was carried out by Luque et al. [9]. In that enumeration, a total of 135 nominal species and 40 unidentified species were reported. These figures stand in contrast to the 259 nominal species and 79 undetermined species documented in the current work, signaling a substantial expansion in knowledge regarding the diversity of Peruvian monogeneans in a mere span of 7 years (Figure 2).

The taxonomic exploration of monogeneans in Peru has primarily relied on morphological data [41,43,49,52,53,63,64,85,106,111,112,113,116,121,125,137,138] with limited integration of molecular methodologies to elucidate phylogenetic relationships among species [34,36,54,57,61,62,107,141,146,149]. As of the present date, molecular characterization has been performed on 51 monogenean species, including 48 species within the Dactylogyridae [34,36,54,57,61,62,107], 2 within the Hexabothriidae [146,149], and 1 within the Monocotylidae [141]. Dactylogyrids infecting siluriform fish have been the most extensively studied species using molecular approaches to comprehend their phylogenetic interrelations [34,36,54,57,61,62,107]. On the other hand, in marine monogeneans, specifically polyopisthocotylean, there have been few studies that have combined both morphological and molecular datasets [146,149]. Integrating these approaches is pivotal for a comprehensive understanding of the intricate taxonomy and phylogenetic associations within these parasites. A more holistic perspective, combining morphological characters and molecular markers, is crucial to delineate species boundaries, resolve evolutionary relationships, and elucidate the co-evolutionary dynamics between monogeneans and their host species, thus advancing our comprehension of their diversity and evolutionary history [197].

A major concern highlighted in this study pertains to the deactivation of two national scientific collections, IMT and CHURP, which previously housed a substantial quantity of type materials and vouchers of monogeneans, resulting in the loss of these specimens. These collections, initially established through the dedication of Peruvian researchers, have gradually dissipated over time with the absence of their founders, resulting in a significant loss of valuable material and exerting a detrimental impact on monogenean research in Peru. A specific example is observed in Oliva & Luque [89], who reported on three unidentified species of Diplectanidae infecting Sciaenidae fish: Diplectanidae gen. sp1. and Diplectanidae gen. sp2. from *C*. *fasciatum*, and Diplectanidae gen. sp3. from *P*. *peruanus*. The voucher material of these species was deposited in the CHURP, but since this collection has been deactivated and its material has been misplaced, an irreparable loss has occurred. Furthermore, Chero et al. [124] described a new *Rhamnocercus* species, *R*. *dominguesi*, in the same hosts studied by Oliva & Luque [89], suggesting that the specimens reported as Diplectanidae gen. sp3. by Oliva & Luque [89] could correspond to those described by Chero et al. [124]. Similarly, Chero et al. [125] described *R*. *fasciatus* in *C*. *fasciata*. This species may possibly be conspecific with either Diplectanidae gen. sp1. or Diplectanidae gen. sp2 from Oliva & Luque [89]. Unfortunately, this claim cannot be verified due to the loss of voucher specimens, resulting in a significant gap in available scientific information on Peruvian monogeneans. A more critical situation emerged with the complete loss of the type material deposited at CHURP for *Euryhaliotrema paralonchuri*, *Rhamnocercoides menticirrhi*, *Rhamnocercus oliveri*, and *R*. *stelliferi*. Chero et al. [124] redescribed *Rh*. *menticirrhi* and designated new type material (neotype and paraneotypes). Nevertheless, the other three species still require the designation of new type material, posing challenges for future research. Additionally, part of the type material for *Anoplocotyloides chorrillensis* (holotype), *Hargicotyle magna* (paratypes), *H*. *paralonchuri* (paratypes), *Pedocotyle annakohnae* (paratypes), and *P*. *bravae* (paratypes) from CHURP has been lost. Fortunately, specimens of these species were deposited in different collections, such as MUSM-HEL, USNM, CHIOC, and CNHE. Furthermore, depositing type material or voucher specimens of parasites in enduring scientific collections is essential for preserving natural history and biological diversity, facilitating current and future research, and supporting integrity and replicability in advancing scientific knowledge [198]. The Scientific Collection of Helminths and Related Invertebrates at the Natural History Museum of the National University of San Marcos (MUSM-HEL) stands as the most significant collection in Peru, is acknowledged internationally, and has been actively contributing since 1960. This collection houses a considerable number of type materials of monogenean parasites, predominantly derived from marine environments. Strengthening this collection hinges on researchers considering it as a primary option for depositing their type materials of helminths.

## 5. Conclusions

The checklist reveals a significant diversity of monogeneans in Peru, with 358 taxa identified, including 270 nominal species and 88 taxa classified at various hierarchical levels. Additionally, the study emphasizes the dominance of certain monogenean families, particularly Dactylogyridae and Diplectanidae, which constitute the majority of recorded species. Lastly, the checklist raises concerns about the loss of scientific collections containing valuable type materials and voucher specimens of monogeneans.

## Figures and Tables

**Figure 1 animals-14-01542-f001:**
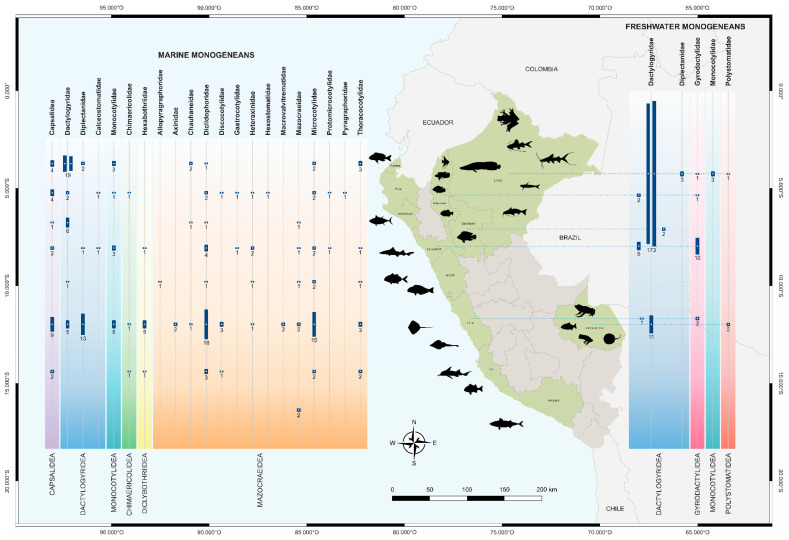
Diversity of monogeneans associated with marine and freshwater fish from Peru.

**Figure 2 animals-14-01542-f002:**
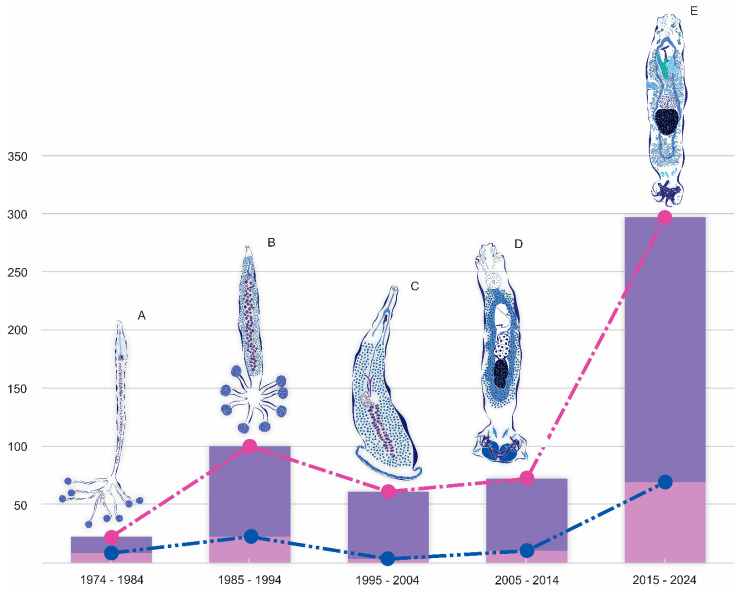
Number of newly described monogenean (blue line) taxa and total records of species (pink line) between 1974 and 2024 in fish of Peru. A: *Hargicotyle peruensis*. B: *Hargicotyle menticirrhi*. C: *Loxura peruensis*. D: *Demidospermus peruvianus*. E: *Brotulella luisahelenae*.

**Table 1 animals-14-01542-t001:** Acronyms used in the checklist.

Acronym	Collections
CHIOC	Helminthological Collection of the Oswaldo Cruz Institute, Brazil.
CHURP	Helminthological Collection of Ricardo Palma University.
CNHE	National Collection of Helminths at the Institute of Biology, National Autonomous University of Mexico, Mexico.
CPMP	Collection of Protozoa and Metazoan Parasites at Federico Villarreal University, Peru.
HPIA	Collection of Parasitic Helminths and Related Invertebrates at the Natural History Museum of Federico Villarreal University.
IMT	Collection of the “Daniel Alcides Carrión” Tropical Medicine Research Institute
INPA	Zoological Collection of the Instituto Nacional de Pesquisas da Amazônia, Manaus, Brazil.
IPCAS	Helminthological Collection of the Institute of Parasitology, České Budějovice, Czech Republic.
LAPYSA	Parasitology and Aquaculum Health Laboratory Collection of the Peruvian Amazon Research Institute, Peru.
NHMUK	Natural History Museum, London, United Kingdom.
MUSM-HEL	Helminthological and Minor Invertebrates Collection of the Museum of Natural History at San Marcos University, Peru.
SMNK	State Museum of Natural History, Karlsruhe, Germany.
UNMSM-LPFSZ	Collection of the Parasitology of Wild Animals and Zoonoses of the Biological Sciences Faculty at San Marcos University, Peru.
USNM	Helminthological Collection of the National Museum of Natural History, Smithsonian Institution, United States.
USNPC	United States National Parasite Collection, United States.

## Data Availability

The data presented in this study are contained within the article.

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
