# Peer review of "An Annotated Checklist of Monogeneans (Platyhelminthes, Monogenea) from Aquatic Vertebrates in Peru: A Review of Diversity, Hosts and Geographical Distribution"

_animals, 2024, doi:10.3390/ani14111542_

Round 1
Reviewer 1 Report
Comments and Suggestions for Authors
The manuscript is a very thorough compilation of not only publications on the monogeneans of Peru, but also data on specimens from scientific collections and some new data, which makes this article very useful for researchers.
I have no major comments on the work.
There are some minor technical corrections:
1. In some cases, where information on findings is repeated, you could include references to articles in general parentheses.
For example,
Lines 792-803 could be more compactly presented as:
Dawestrema cycloancistrium Price & Nowlin, 1967
Arapaima gigas; gills; freshwater; Loreto (CHURP 701) [15].
Arapaima gigas; gills; freshwater, Loreto (3º49'S, 73º19'W) [71].
Arapaima gigas; gills; freshwater; Loreto [73 74, 75, 76, 77, 79].
Arapaima gigas; gills; freshwater, Loreto (05°54'S, 76°05'W) (CHIOC 38.656 a−c, 38657 800 a−c) [78].
Similarly, lines 989-992, 1618-1623, 1668-1670.
Check where else in similar cases the data can be presented more compactly.
2. Figure 1 – It would be useful to present the scale of the blue lines by number of species.
3. Figure 2. Number of total records (pink line) and new monogenean (blue line) taxa between 1974 2991 and 2024 in fishes of Peru.
The legend of the figure is not very clear.
New monogeneans - are these species new to science or are they first recorded in Peru?
What is "total records" - is it the number of species?
If it is the number of species that were known at the end of the period, why did the number of species decrease from 1995 to 2014?
Nor does it explain why these species are shown in the figure? Were these new species described during this period? If so, it should be explained.
It would be better to show exactly how many species were known at the end of each period, and how many of these were recorded for the first time during the period.
To show progress in the study of monogeneans in Peru, it is probably appropriate to show the number of species found for the first time in a given period and the total number of species known at the end of the period.
Author Response
Response to Reviewer 1 Comments
The manuscript is a very thorough compilation of not only publications on the monogeneans of Peru, but also data on specimens from scientific collections and some new data, which makes this article very useful for researchers.
I have no major comments on the work.
There are some minor technical corrections:
- In some cases, where information on findings is repeated, you could include references to articles in general parentheses.
For example,
Lines 792-803 could be more compactly presented as:
Dawestrema cycloancistrium Price & Nowlin, 1967
Arapaima gigas; gills; freshwater; Loreto (CHURP 701) [15].
Arapaima gigas; gills; freshwater, Loreto (3º49'S, 73º19'W) [71].
Arapaima gigas; gills; freshwater; Loreto [73 74, 75, 76, 77, 79].
Arapaima gigas; gills; freshwater, Loreto (05°54'S, 76°05'W) (CHIOC 38.656 a−c, 38657 800 a−c) [78].
Similarly, lines 989-992, 1618-1623, 1668-1670.
Check where else in similar cases the data can be presented more compactly.
-Ok. It was made.
- Figure 1 – It would be useful to present the scale of the blue lines by number of species.
-Ok. It was made.
- Figure 2. Number of total records (pink line) and new monogenean (blue line) taxa between 1974 2991 and 2024 in fishes of Peru.
The legend of the figure is not very clear.
-Ok. It was modified.
New monogeneans - are these species new to science or are they first recorded in Peru?
What is "total records" - is it the number of species?
If it is the number of species that were known at the end of the period, why did the number of species decrease from 1995 to 2014?
Nor does it explain why these species are shown in the figure? Were these new species described during this period? If so, it should be explained.
It would be better to show exactly how many species were known at the end of each period, and how many of these were recorded for the first time during the period.
To show progress in the study of monogeneans in Peru, it is probably appropriate to show the number of species found for the first time in a given period and the total number of species known at the end of the period.
-Ok. It was modified.
Reviewer 2 Report
Comments and Suggestions for Authors
Please see the attached file.

Moderate editing of English language required
Author Response
Response to Reviewer 2 Comments
I reviewed the manuscript (MS) “An annotated checklist of monogeneans (Platyhelminthes, Monogenea) from aquatic vertebrates in Peru: a review of diversity, hosts and geographical distribution” by Luis Angel Santillan and colleagues. The MS reports for the first time an extensive checklist reporting the status of monogenean trematodes in aquatic organisms from Peru. In general, the MS could be considered as complete report of many species infecting fish and other aquatic vertebrates, including also unpublished data useful to better describe the dynamics of these parasites in the aquatic ecosystem. The MS presents some minor lacks, and, in some parts, some improvements should be added. However, the study design results well structured, even if some comments can help the authors to better report their results. Moreover, the Discussion section well support the obtained results, resulting useful to understand the results. Conclusion should be reduced and better written.
However, the English should be reviewed by native speaker before the publication.
-Ok. It was made.
Some suggestions are reported below.
Lines 30 – 31: Please rephrase “…and where these parasites are found geographically.” to “…and geographical distribution.”, as reported in line 99.
-Ok. It was made.
Line 31: authors in line 29 – 30 reported that parasite from collection have been examined. Here they reported “found”. I suggest them to change “found” to “reported”.
-Ok. It was made.
Line 34 – 36: Please rephrase these sentences.
-Ok. It was made.
Lines 37 – 64: Please reduce the abstract according to the instruction for the authors.
-Ok. It was made.
Line 73 – 74: Please change “…certain species, particularly members of the Polystomatidae family...” to “…species belonging to the family Polystomatidae…”.
-Ok. It was made.
Lines 75 – 77: Please rephrase this sentence because monogenean are characterized by a direct life cycle.
-Ok. It was made.
Line 81: Please change “constitute” to “represent”.
-Ok. It was made.
Lines 84 – 85: Please add the name of the authority (just the first time) as already done for other organisms.
-Ok. It was made.
Line 88: Please change “of” to “from”.
-Ok. It was made.
Lines 101 – 102: Please rephrase this sentence.
-Ok. It was made.
Line 105: Please change “search” to “analysis” or “evaluation”.
-Ok. It was made.
Lines 106 – 107: Please add in brackets all databases included in the study, deleting “etc.”
-Ok. It was made.
Lines 107 – 108: “…following international and national collections.” Please add here the list or change as “2) analysis conducted on the database of the international and national collections are reported below;…”.
-Ok. It was made.
Line 113: Please delete “…in monogenean names...”.
-Ok. It was made.
Line 117 – 118: Please change “In case of new species, the parentheses indicate the type locality, type host, and original reference.” to “In case of new species, the type locality, type host, and original reference are reported in brackets.”
-Ok. It was made.
Line 120: Please change “features” to “reports”.
-Ok. It was made.
Lines 122 – 145: I suggest the authors to report these data in a table.
-Ok. It was made.
Line 147: Please change “compiled” to “reported”.
-Ok. It was made.
Line 152: Please remove “intensively”.
-Ok. It was made.
Line 153 – 155: In my opinion, this part should be moved to the discussion section.
-Ok. It was made.
Line 156: Please add space after the brackets.
-Ok. It was made.
Line 156: Please uniform the nomenclature, reporting the authority without brackets, as already done in other parts of the MS.
-Ok. It was made.
Lines 162 – 163: It is not so easy to understand that sentence, please rephrase, mainly in the last part. What the authors mean with “…with 9 and 7 respectively, are the most diverse genera”? Please explain.
-Ok. It was made.
Line 198: Please change “gill” to “gills”. Please carefully check that also in other parts of the MS.
-Ok. It was made.
Line 376: Please add space before “gills”.
-Ok. It was made.
Line 677: Please remove the italics from “sp.”. Please check also in other parts of the MS (Line 773, 2525, 2656, 2757).
-Ok. It was made.
Line 1285: Please add space before “by”.
-Ok. It was made.
Line 1693: What the authors mean with “body washings”? Maybe scraping and collecting the material from the skin? Please explain.
-Ok. It was made.
Lines 2261 – 2265: Please check the italics in the scientific names.
-Ok. It was made.
Line 2837: Please add “.” after “sp”.
-Ok. It was made.
Lines 2968 -2969: Please move “worldwide” at the end of the sentence.
-Ok. It was made.
Line 2970: Please change “represented” to “reported”.
-Ok. It was made.
Figure1: I suggest the authors to write in bold the family names to highlight them.
-Ok. It was made.
Figure2: Please improve the quality of the image.
-Ok. It was made.
Line 3007: Please change “augmented” to “increased”. In general, please rephrase this sentenced.
-Ok. It was made.
Line 3017: Please report the references in the text according to the instruction. Please check also in other parts of the MS.
-Ok. It was made.
Lines 3019 – 3021: Please rephrase this sentence.
-Ok. It was made.
Line 3028: Please report “...this parasites.” Or “… this organisms.”
-Ok. It was made.
Line 3044 – 3049: These sentences could be considered a speculation more than a data because the authors didn’t have access to the original data and description, even if the authors reported this gap in the next sentence. Please rephrase this sentence highlighting that also here.
-Ok. It was made.
Lines 3053 – 3054: Please check “redescripción”.
-Ok. It was made.
Lines 3060 – 3061: It is not so easy to deposit more than one specimen in different collection, often the description has been done on few numbers of specimen. I suggest to delete this sentence.
-Ok. It was made.
Lines 3073 – 3097: The conclusion section is reported as a second abstract without material and methods. I suggest the authors to change the conclusion section as a single paragraph, deleting “primarily, secondly, thirdly…..”, reporting their conclusion followed by their final consideration. At the same time, I suggest them to reduce this section.
-Ok. It was corrected.
Reviewer 3 Report
Comments and Suggestions for Authors
The work contains a list of monogeneans from fish and amphibians in Peru. Checklists are of great value, so other researchers will certainly be interested in this work.
I am of an opinion that the article fits into scope of Animals and could be published after major corrections.
Comments:
1. Abstract: the abstract must state how many valid species were found; the number 358 may be misleading. Please change this sentence, maybe to "...found 358 species (including 270 valid) ??
2. lines 34-35: „Some 33 fish species harbor more parasites than others.” – not clear ?
3. line 39: “… invertebrates like copepods and cephalopods …” – “…invertebrates like copepods, isopods and cephalopods…” - ?
4. lines 45: “and the inclusion of additional freshly collected specimens” - ??, no information in the "Materials and methods" and "Results" chapters that freshly samples were also collected (what and how many species of fish, what methods were used, etc.)?
5. Keywords: the word "amphibians" should also be added here.
6. lines 54-55: entire scientific names (with authors and dates) are already in the checklist, so they are unnecessary here.
7. Introduction: some entire scientific names of host (with authors and dates) are already in the checklist (e.g. ).
8. Lines 123-145: IMT is missing.
9. „Parasite-host list” – „Remarks”: for appropriate species, please add "valid”.
10. Line175: „stomatch;” - ?
11. lines 214, 264, 596, etc, etc: “(new geographical record) (present study)” – not clear, it is not known what "present study" is meant; please see also comment 3.
12. line 773, etc.: “sp.” should be italicized.
13. line 775, etc, etc: why GPS wasn't always provided ?
14. line 796, etc, etc: please check GPS - sometimes the degree symbol is underlined.
15. Lines 557-558, 918, etc, etc.: Inconsistent citation styles - sometimes there is "&", sometimes "and".
16. Line 1805, etc, etc.: - why is there sometimes "Note" instead of "Remarks"?
17. Line 1852: „Remark”- remarks
18. Line 1223: „Mizelle & Kritsky, 1969” – is this a quote or a species author?; in my opinion this is citations, that's why it should be "Mizelle and Kritsky [...]".
19. Lines 2349, 2351, etc.: - please complete the specimen numbers.
Comments on the Quality of English LanguageMinor editing of English language required.
Author Response
Response to Reviewer 3 Comments
- Abstract: the abstract must state how many valid species were found; the number 358 may be misleading. Please change this sentence, maybe to "...found 358 species (including 270 valid) ??
-Ok. It was made.
- lines 34-35: „Some 33 fish species harbor more parasites than others.” – not clear ?
-Ok. It was corrected.
- line 39: “… invertebrates like copepods and cephalopods …” – “…invertebrates like copepods, isopods and cephalopods…” - ?
-Ok. It was made.
- lines 45: “and the inclusion of additional freshly collected specimens” - ??, no information in the "Materials and methods" and "Results" chapters that freshly samples were also collected (what and how many species of fish, what methods were used, etc.)?
-Ok. It was added.
- 5. Keywords: the word "amphibians" should also be added here.
-Ok. It was made.
- lines 54-55: entire scientific names (with authors and dates) are already in the checklist, so they are unnecessary here.
- According to the rules of the journal and the International Code of Zoological Nomenclature, scientific names must be accompanied by authors and year when they are mentioned for the first time.
- Introduction: some entire scientific names of host (with authors and dates) are already in the checklist (e.g. ).
- According to the rules of the journal and the International Code of Zoological Nomenclature, scientific names must be accompanied by authors and year when they are mentioned for the first time.
- Lines 123-145: IMT is missing.
-Ok. It was made.
- „Parasite-host list” – „Remarks”: for appropriate species, please add "valid”.
- We prefer not to use the term 'valid species' to avoid conflicts, as we have not had access to all the type or voucher materials of the monogenean species registered in Peru.
- 10. Line175: „stomatch;” - ?
-Ok. It was corrected.
- lines 214, 264, 596, etc, etc: “(new geographical record) (present study)” – not clear, it is not known what "present study" is meant; please see also comment 3.
- The mention of '(new geographical record) (present study)' refers to species found in Peruvian collections that were previously unreported in formal publications. Our work identified and documented these species. When a species was not previously recorded in Peru, we marked it as a new geographical record, and the reference is from the present study.
- line 773, etc.: “sp.” should be italicized.
-Ok. It was corrected.
- line 775, etc, etc: why GPS wasn't always provided ?
-Not all articles provided GPS data.
- line 796, etc, etc: please check GPS - sometimes the degree symbol is underlined.
-Ok. It was corrected.
- Lines 557-558, 918, etc, etc.: Inconsistent citation styles - sometimes there is "&", sometimes "and".
-Ok. It was corrected.
- Line 1805, etc, etc.: - why is there sometimes "Note" instead of "Remarks"?
-A 'Note' refers to additional data or information, while 'Remarks' pertain to taxonomic issues.
- Line 1852: „Remark”- remarks
-Ok. It was corrected.
- Line 1223: „Mizelle & Kritsky, 1969” – is this a quote or a species author?; in my opinion this is citations, that's why it should be "Mizelle and Kritsky [...]".
-Ok. It was corrected.
- Lines 2349, 2351, etc.: - please complete the specimen numbers.
-Ok. It was corrected.
Round 2
Reviewer 3 Report
Comments and Suggestions for Authors
The authors did not addressed (did not understand) some of my comments, therefore the work still requires changes.
My comments:
6. lines 54-55: entire scientific names (with authors and dates) are already in the checklist, so they are unnecessary here.
- According to the rules of the journal and the International Code of Zoological Nomenclature, scientific names must be accompanied by authors and year when they are mentioned for the first time.
My new comment: it has nothing to do with the code. The entire species name is given only once in the text, here are examples:
The name „Mesopolystoma samiriensis Vaucher, 1981” was repeated three times (lines: 54, 2364, 2930).
The name „Wetapolystoma almae Gray, 1993” was repeated three times (lines 55, 2374, 2931).
7. Introduction: some entire scientific names of host (with authors and dates) are already in the checklist (e.g. ).
- According to the rules of the journal and the International Code of Zoological Nomenclature, scientific names must be accompanied by authors and year when they are mentioned for the first time.
My new comment: it has nothing to do with the code. The entire species name is given only once in the text, here is an example: lines: 83-84, 2722 - “Seriolella violacea Guichenot, 1848”.
9. „Parasite-host list” – „Remarks”: for appropriate species, please add "valid”.
- We prefer not to use the term 'valid species' to avoid conflicts, as we have not had access to all the type or voucher materials of the monogenean species registered in Peru.
My new comment: the term "valid species" is a term used in taxonomy. A cheklist without information about which species are "valid" and which "do not have valid status" makes no sense and is worthless. Cheklists are made by specialists and they should be able to verify the status of species. This does not require carrying out a systematic review of species (e.g. analyzes of museum specimens, voucher materials), but only verifying the status on the basis of the principles of the International Code of Zoological Nomenclature.
In the abstract, the authors use the term "valid" ?
Author Response
Dear reviewer, thank you very much for your comments. We have accepted all the changes you indicated in your review.
